# TFIP11 promotes replication fork reversal to preserve genome stability

Junliang Chen[1,2,3,6], Mingjie Wu[4,6], Yulan Yang[2,6], Chunyan Ruan[2,6], Yi Luo[2], Lizhi Song [2], Ting Wu[2], Jun Huang [1,2], Bing Yang [2] & Ting Liu [1,5] ✉

Replication fork reversal, a critical protective mechanism against replication stress in higher eukaryotic cells, is orchestrated via a series of coordinated enzymatic reactions. The Bloom syndrome gene product, BLM, a member of the highly conserved RecQ helicase family, is implicated in this process, yet its precise regulation and role remain poorly understood. In this study, we demonstrate that the GCFC domain-containing protein TFIP11 forms a complex with the BLM helicase. TFIP11 exhibits a preference for binding to DNA substrates that mimic the structure generated at stalled replication forks. Loss of either TFIP11 or BLM leads to the accumulation of the other protein at stalled forks. This abnormal accumulation, in turn, impairs RAD51-mediated fork reversal and slowing, sensitizes cells to replication stress-inducing agents, and enhances chromosomal instability. These findings reveal a previously unidentified regulatory mechanism that modulates the activities of BLM and RAD51 at stalled forks, thereby impacting genome integrity.

Accurate and complete replication of the genetic material is vital for maintaining genomic integrity[1,2]. However, DNA replication faces constant challenges from various sources of stress, both endogenous and exogenous, such as unusual secondary DNA structures, tightly DNA-bound proteins, and DNA lesions[1,2]. Failure to properly address these challenges can result in replication fork stalling and genomic instability, ultimately contributing to tumorigenesis[1,2]. A conserved cellular response to replication stress is replication fork reversal, which has been shown to be essential for stabilizing stalled replication forks[3–6]. This process involves converting the three-armed replication fork into a Holliday junction structure, often referred to as the chicken-foot structure. This conversion occurs by pairing the extruded nascent strands to form a fourth regressed arm[3–6]. We recently proposed that replication fork reversal is carried out via a two-step mechanism[7,8]. In the first step, DNA translocases like SMARCAL1, ZRANB3, and HLTF, initiate limited fork reversal, creating positive superhelical strain in the newly replicated sister chromatids[5,9–12]. In the second step, the DNA

topoisomerase TOP2A facilitates extensive replication fork reversal. It does so by relieving topological barriers and recruiting the SUMO-targeted DNA translocase PICH to stalled replication forks[7].

The RAD51 recombinase is an essential factor of homologous recombination (HR) that plays crucial roles in the repair of DNA double-strand breaks (DSBs) and in safeguarding genome integrity[13]. It is also required for protecting stalled replication forks from nuclease-catalyzed degradation of nascent strands[14,15]. The HR and fork protection functions of RAD51 are strictly dependent on the mediator protein BRCA2, which assists RAD51 in replacing RPA to form a stable nucleofilament[14,16]. Apart from its role in these events, accumulating lines of evidence indicate that RAD51 is also involved in promoting replication fork reversal in response to replication stress[3]. It has been proposed that RAD51 acts together with DNA translocases to catalyze the initiate fork reversal step[3,7]. Unexpectedly, RAD51-mediated replication fork reversal occurs in a manner independent of BRCA2[17]. To date, precisely how RAD51

¹Zhejiang Provincial Key Laboratory of Geriatrics and Geriatrics Institute of Zhejiang Province, Affiliated Zhejiang Hospital, Zhejiang University School of Medicine, 310058 Hangzhou, China. ²The MOE Key Laboratory of Biosystems Homeostasis & Protection and Zhejiang Provincial Key Laboratory of Cancer Molecular Cell Biology, Life Sciences Institute, Zhejiang University, 310058 Hangzhou, China. ³Center for Life Sciences, Shaoxing Institute, Zhejiang University, 321000 Shaoxing, China. ⁴The Trauma Center, The First Affiliated Hospital, Zhejiang University School of Medicine, 310058 Hangzhou, China. ⁵Department of Cell Biology, Zhejiang University School of Medicine, 310058 Hangzhou, China. ⁶These authors contributed equally: Junliang Chen, Mingjie Wu, Yulan Yang, Chunyan Ruan. ✉e-mail: liuting518@zju.edu.cn

activity at stalled forks is controlled in cells remains poorly understood.

BLM is a member of the evolutionarily conserved RecQ DNA helicase family[18]. Loss-of-function mutations in the BLM gene cause Bloom syndrome, a rare and recessive genetic disorder associated with an increased risk of various types of cancer[18,19]. In mammalian cells, BLM forms the BTR (BLM-TOP3A-RMI1/2) complex by associating with topoisomerase IIIalpha (TOP3A), RMI1 (BLAP75), and RMI2 (BLAP18)[19–27]. The well-established function of this complex is to facilitate the dissolution of double Holliday junctions, late DNA intermediates generated during HR, resulting in non-crossover recombination products[28–35]. This activity explains the observed increase in sister chromatid exchange frequency in Bloom syndrome cells[18,25,36–39]. Beyond its role in late HR events, BLM has been implicated in early stages of the HR pathway. Biochemical evidence demonstrates that BLM can displace the RAD51 recombinase from single-stranded DNA (ssDNA) in an ATPase-dependent manner[40–43]. Additionally, BLM acts epistatically with the DNA2 nuclease to promote long-range resection of DSB ends[44–46]. Besides its critical roles in HR, BLM has been implicated in facilitating restart of stalled replication forks via promoting fork reversal[47,48]. However, the precise mechanisms by which BLM participates in this process and how its activities at stalled forks are regulated in response to replication stress remain largely unknown.

The GCFC domain-containing protein TFIP11 (also known as TIP39) was initially identified as a tuftelin binding partner through yeast two-hybrid screening[49]. Despite implications of TFIP11 and its homologs in various biological processes, including RNA splicing[50–52], non-homologous end-joining (NHEJ)-mediated DSB repair[53], and telomere metabolism[53], the precise biochemical mechanisms remain to be defined. Additionally, the potential involvement of TFIP11 in the cellular response to replication stress has yet to be explored. In this study, we reveal that TFIP11 plays a crucial role in replication fork reversal by modulating the activities of BLM and RAD51 at stalled forks. Our findings offer valuable insights into the molecular mechanisms governing replication fork reversal and the stability of stalled forks.

## Results

### TFIP11 interacts with the BLM complex

To elucidate the biological function of TFIP11, we established a stable HEK293T cell line expressing wild-type TFIP11 with triple-epitope tags (S-protein, Flag, and streptavidin-binding peptide) for the isolation of TFIP11-associated proteins. Following a tandem affinity purification (TAP) scheme, proteins copurifying with TFIP11 were identified through mass spectrometric analysis (Fig. 1A, B). Proteins with a peptide-spectrum match (PSM) ≥ 4 were considered high-confidence interacting proteins (Supplementary Data 1). Gene ontology (GO) term analysis revealed that TFIP11 interactors are predominantly involved in RNA splicing and telomere organization, aligning with findings from previous studies[50–53] (Supplementary Fig. 1A). Notably, the analysis detected the presence of BLM and several established BLM-associated factors, including TOP3A, RMI1, RMI2, RIF1, FANCI, and FANCD2, among the prominent TFIP11-associated proteins (Fig. 1C and Supplementary Fig. 1A).

To validate our TAP-MS results, we carried out co-immunoprecipitation experiments to confirm the interaction between endogenous TFIP11 and the BLM protein complex. HeLa cell extracts were prepared and subjected to immunoprecipitation assays with either control IgG or anti-TFIP11 antibody (Fig. 1D). Western blot analysis revealed that BLM, TOP3A, RMI1, and RIF1 were clearly detected in the immunoprecipitations obtained with

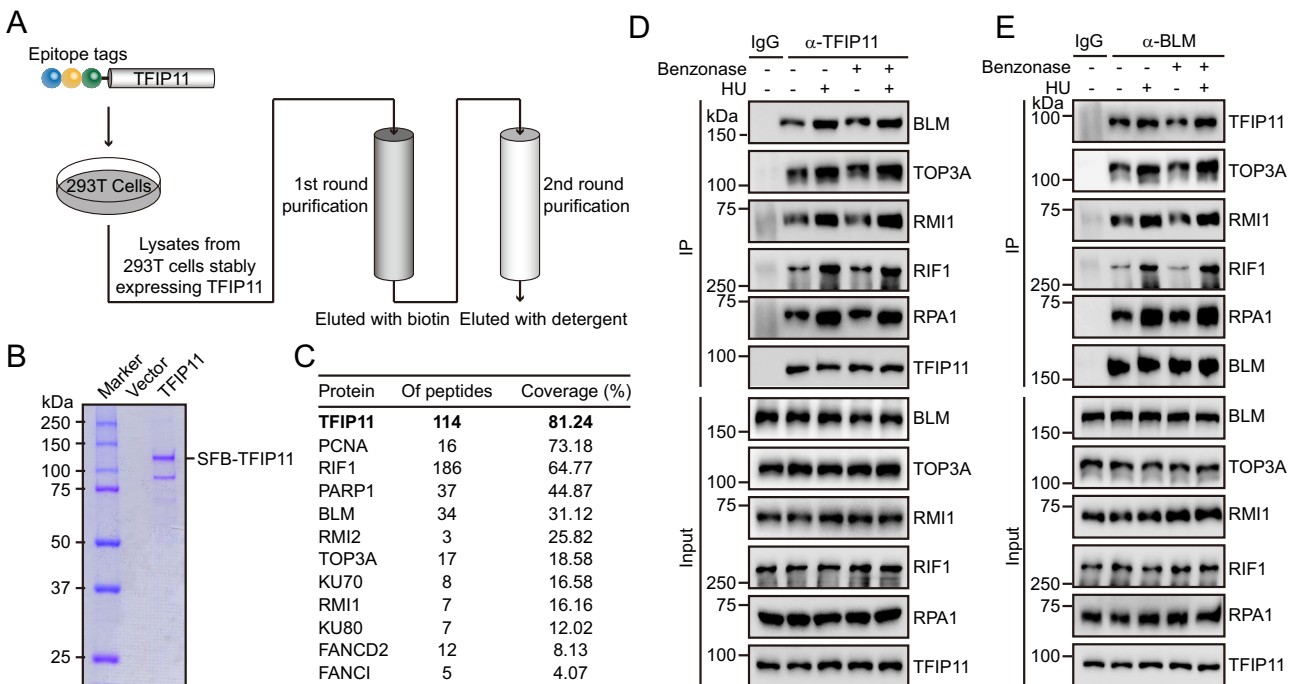

**Fig. 1 | TFIP11 interacts with the BLM complex. A** Tandem affinity purification (TAP) procedure. **B, C** HEK293T cells stably expressing either vector control or SFB-tagged TFIP11 were harvested and lysed with NETN buffer for 30 min on ice. Crude lysates were centrifuged at 4 °C, 14,000 × g for 8 min and the resulting supernatants were incubated with streptavidin beads at 4 °C for 2 h. After three washes, immunocomplexes were eluted with NETN buffer containing 1 mg/ml biotin. Elutes were then incubated with S-protein beads at 4 °C for 2 h. The final eluates were resolved by SDS-PAGE, stained with Coomassie blue (**B**) or subjected to mass spectrometric analysis (**C**). **D, E** Association of endogenous TFIP11 with the BLM complex. HeLa cells were mock treated or treated with 4 mM HU for 3 h. The cells were then lysed with NETN buffer in the presence or absence of Benzonase. The resulting cell lysates were incubated with protein A agarose beads conjugated to either anti-TFIP11 antibodies (**D**) or anti-BLM antibodies (**E**), and then subjected to Western blotting. Source data are provided as a Source Data file.

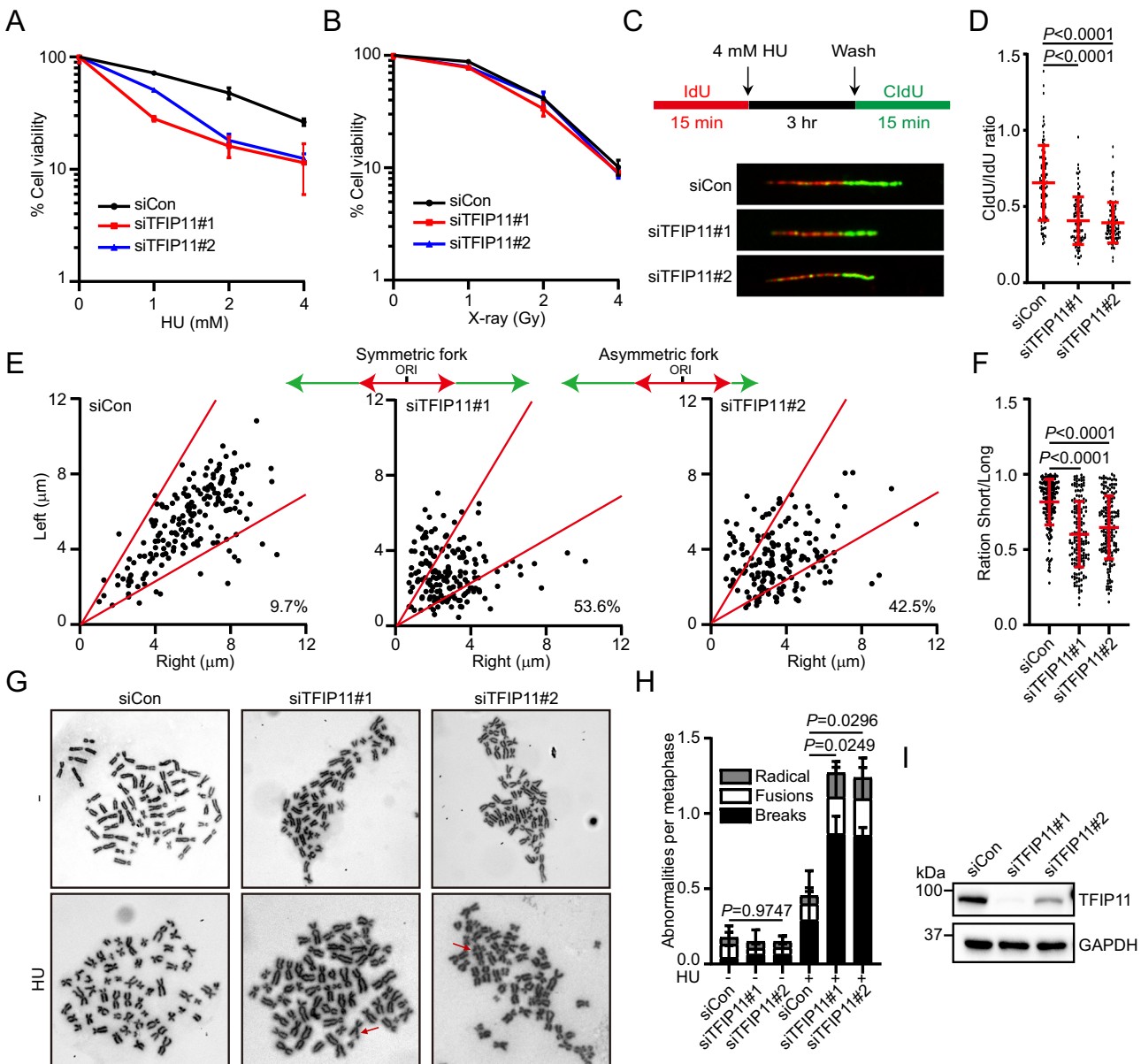

**Fig. 2 | TFIP11 is involved in replication stress response. A, B** Clonogenic survival assays in TFIP11-depleted U2OS cells after HU or X-ray treatment. Results represent means ± SD from three independent experiments. **C** Top: schematic representation of the DNA fiber assay. HeLa cells were labeled with IdU (red) for 15 min, challenged with 4 mM HU for 3 h, and then labeled with CldU (green) for 15 min. Bottom: representative fiber images for each sample. **D** Dot plot of CldU/IdU ratios for individual replication forks. The data represent the mean ± SD from three independent experiments. From left, $n = 101, 101, 101$ fibers. $P$ values were derived from a one-way ANOVA with Tukey's multiple comparisons test. **E, F** HeLa cells were labeled with IdU (red) for 15 min, challenged with 2 mM HU for 2 h, and then labeled with CldU (green) for 15 min. CldU lengths from left and right moving forks from the same origin were measured (**E**). In (**F**), the ratio of the sister fork lengths is plotted. The data represent the mean ± SD from three independent experiments. From left, $n = 185, 166, 167$ fibers. $P$ values were derived from a one-way ANOVA with Tukey's multiple comparisons test. **G–I** Representative images of metaphase spreads prepared from HU-treated HeLa cells transfected with indicated siRNAs. Chromosomal aberrations are marked by arrows (**G**). Quantification of chromosomal aberrations (**H**). Data are presented as mean ± SD of 50 metaphases per indicated condition. $P$ values were derived from a one-way ANOVA with Tukey's multiple comparisons test. Knockdown efficiency was confirmed (**I**). Source data are provided as a Source Data file.

the anti-TFIP11 antibody but not with the control IgG (Fig. 1D). Reciprocal co-immunoprecipitation experiments confirmed the interaction, with endogenous TFIP11 readily co-immunoprecipitated by the BLM-specific antibody (Fig. 1E). Significantly, treatment with the ribonucleotide reductase inhibitor hydroxyurea (HU) enhanced the interaction between TFIP11 and the BLM protein complex (Fig. 1D, E). The TFIP11-BLM interaction was resistant to benzonase treatment, ruling out the possibility of nucleic acids bridging the association (Fig. 1D, E). Collectively, these results suggest that TFIP11 associates with the BLM complex.

## TFIP11 participates in cellular response to replication stress

BLM plays an important roles in cellular response to DNA damage and replication stress. The observation that TFIP11 interacts with the BLM complex prompted us to test whether TFIP11 deficiency would hypersensitize cells to agents that induce replication stress or DNA damage. As shown in Supplementary Fig. 1B, TFIP11 depletion did not significantly affect cell proliferation and cell cycle distribution. Interestingly, downregulation of TFIP11 rendered cells more sensitive to HU treatment (Fig. 2A). By contrast, TFIP11-depleted cells did not show sensitivity to X-ray radiation (Fig. 2B). These results suggest that TFIP11

might be specifically involved in cellular response to replication stress. We next examined whether TFIP11 may be required for restart of stalled replication forks by use of the DNA combing method. Accordingly, HeLa cells were first pulse-labeled with modified thymidine analog iododeoxyuridine (IdU), incubated with HU to induce replication fork stalling, and were subsequently released into media containing a second thymidine analog chlorodeoxyuridine (CldU). As shown in Fig. 2C, D, compromising TFIP11 expression resulted in a substantial decrease in the ratio of CldU to IdU track lengths. Similar findings were also observed in U2OS cells (Supplementary Fig. 1C–E). These results indicate that TFIP11 inactivation hampered the restart of stalled replication forks.

Ensuring the timely and efficient restart of stalled forks is vital for maintaining replication fork stability. To assess the stability of ongoing replication forks in response to replication stress, we quantified the symmetry of sister replication forks initiating from the same replication origin in TFIP11-depleted cells. As shown in Fig. 2E, F, knockdown of TFIP11 resulted in a substantial increase in sister fork asymmetry. In accordance with these findings, depletion of TFIP11 significantly elevated the frequency of fork stalling and chromosomal aberrations (Fig. 2G–I and Supplementary Fig. 1F). Taken together, these results indicate that TFIP11 plays a crucial role in safeguarding genome stability after replication stress.

## TFIP11 accumulates at stalled replication forks

To determine whether TFIP11 has a direct role in the replication stress response, we carried out isolation of proteins on nascent DNA (iPOND) assays to examine the association of TFIP11 with stalled replication forks in vivo. As shown in Fig. 3A, PCNA levels at the replication forks were substantially reduced following HU treatment, consistent with previous reports[54]. Strikingly, similar to RAD51, BLM and RMI1[55], TFIP11 was significantly enriched on nascent DNA upon HU treatment, and this enrichment was eliminated after a thymidine chase, indicating that TFIP11 may specifically accumulate within the vicinity of stalled replication forks (Fig. 3A). In support of this, a portion of TFIP11 translocated from the nuclear speckles (where TFIP11 partially colocalizes with the splicing factor snRNP70[50–53]) to stalled forks and colocalized with RPA (Supplementary Fig. 2A, B). To further verify the association of TFIP11 at stalled forks, we employed a proximity ligation assay (PLA)-based approach that measures the association of proteins to nascent DNA. In this assay, HeLa cells stably expressing HA-Flag-tagged TFIP11 were labeled with EdU for 15 min, and were subsequently treated with 4 mM HU for 3 h. Biotin was then conjugated to EdU by click chemistry and PLA was conducted to detect protein binding to biotin-labeled nascent DNA (Fig. 3B). As shown in Fig. 3C, D, HA-Flag-TFIP11 is enriched at nascent DNA upon replication stress. We further used the PLA assay to assess the presence of endogenous TFIP11 at EdU-labeled stalled replication forks. Since our home-made anti-TFIP11 antibody failed to stain endogenous TFIP11 in immunofluorescence staining experiments, we resorted to the CRISPR/Cas9-directed recombinant adeno-associated virus (rAAV)-mediated gene targeting approach to fuse an S-Flag (SF) tag to the N terminus of the endogenous TFIP11 gene (Fig. 3E). DNA sequencing confirmed that the tag was correctly targeted, and expression of SF-TFIP11 protein was also verified by immunoblotting analysis (Fig. 3F). As shown in Fig. 3G–I, cells treated with HU displayed a substantial increase in the number of TFIP11/biotin PLA foci. Taken together, these results suggest that TFIP11 is recruited to stressed replication forks.

## The GCFC domain targets TFIP11 to stalled replication forks

TFIP11 is a multi-domain protein and contains a TIP-N (Tuftelin interacting protein N-terminal) domain followed by a G-patch domain and a GCFC (GC-rich sequence DNA-binding factor-like) domain (Fig. 4A). To identify the region(s) on TFIP11 that may be responsible for its recruitment to stalled replication forks, we generated three TFIP11 deletion mutants, and monitored the recruitment of each of these deletion mutants after replication stress. As shown in Fig. 4B–D and Supplementary Fig. 3A, whereas wild-type TFIP11 and the other mutants retained the ability to associate with nascent DNA at stalled replication forks, the mutant lacking the GCFC domain failed to do so, indicating that the GCFC domain of TFIP11 is indispensable for its recruitment to stalled forks. Strikingly, the GCFC domain of TFIP11 is highly conserved during evolution, and is widely present in eukaryotes from yeast to humans (>64% amino acid similarity between species, Supplementary Fig. 3B), indicating that it might carry out an important function of TFIP11.

## TFIP11 binds to a DNA substrate that mimics the structure generated at stalled forks

To gain insight into the molecular mechanism by which TFIP11 is recruited to stalled forks, we first examined whether TFIP11 recruitment to stalled forks may be dependent on the ATM/ATR kinase pathways. As shown in Supplementary Fig. 3C–E, inhibition of ATM or ATR did not noticeably affect TFIP11 accumulation at stalled forks. Moreover, there was no notable change in the localization of TFIP11 to stalled forks using a PARP1 inhibitor (Supplementary Fig. 3C–E). These findings, together with the observations that the GCFC domain-containing protein possesses putative DNA-binding ability, and is required for TFIP11 recruitment to stalled replication forks, raised the intriguing possibility that TFIP11 might be recruited to stalled forks via direct interaction with DNA. To test this hypothesis, we expressed and purified His-SUMO-tagged wild-type TFIP11 proteins from *E. coli*, and performed electromobility shift assays using a variety of DNA substrates. As shown in Fig. 4E–H, TFIP11 bound strongly to splayed arm, but not to double-stranded DNA (dsDNA) or single-stranded DNA (ssDNA). Strikingly, the GCFC-domain deletion mutant, which does not accumulate at stalled replication forks, exhibited a lack of binding to the splayed arm under the same experimental conditions (Fig. 4I). These results collectively suggest that TFIP11 is a structure-specific DNA-binding protein with a preference for a DNA substrate that mimics stalled replication forks, and that this DNA-binding activity is essential for its accumulation at stalled forks in cells.

## TFIP11 promotes replication fork reversal

Previous studies have shown that replication fork reversal is a global cellular response to replication stress in higher eukaryotic cells[3]. The observations described above led us to hypothesize that TFIP11 may maintain stalled fork stability by promoting replication fork reversal. To test this hypothesis, we applied the electron microscopy approach to monitor the formation of reversed forks in control and TFIP11-depleted cells. As shown in Fig. 5A, B and Supplementary Fig. 4A, TFIP11 depletion resulted in approximately a 2-fold reduction in the percentage of HU-induced reversed forks.

Annealing of the extruded nascent DNA strands during replication fork reversal produces ssDNA end on the regressed arm, which could be detected by the native 5-bromo-2′-deoxyuridine (BrdU) immunofluorescence assay[56] (Fig. 5C). Interestingly, in agreement with the pivotal role of TFIP11 in promoting replication fork reversal, cells depleted of TFIP11 exhibited a marked reduction in HU-induced BrdU foci formation (Fig. 5D, E and Supplementary Fig. 4B).

Fork reversal actively restrains replication fork progression upon replication stress[12,57,58]. Given that TFIP11 is required for efficient fork reversal, we speculated that knockdown of TFIP11 would compromise slowing of replication stress-induced fork progression. To test this hypothesis, we pulse-labeled HeLa cells with IdU followed by labeling with CldU in the presence of low concentrations of HU, and then measured the CldU/IdU ratio, which reflects DNA synthesis under replication stress (Fig. 5F). As expected, depletion of TFIP11 resulted in a substantial increase in the CldU/IdU ratio, suggesting that TFIP11 is necessary for replication stress-induced fork slowing (Fig. 5F–H). More

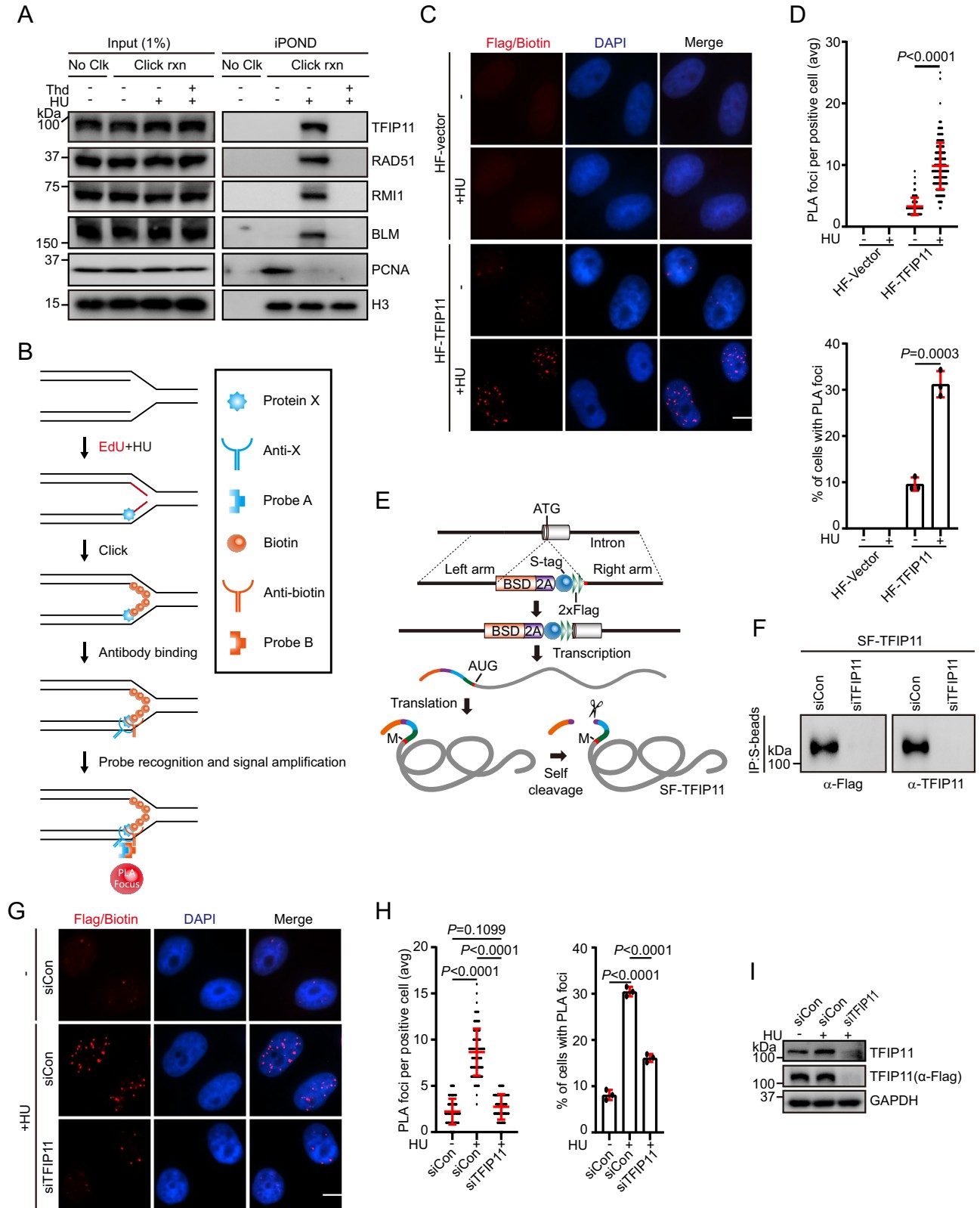

importantly, the defects in fork reversal and slowing associated with TFIP11 deficiency could be reversed by re-expression of siRNA-resistant wild-type TFIP11 but not by re-expression of its GCFC-domain deletion mutant (Fig. 5D–H), indicating that the DNA-binding activity of TFIP11 is critical for its function in promoting reversal fork reversal and slowing following replication stress. In line with these findings, wild-type TFIP11, but not the TFIP11-D3 mutant, was able to

alleviate the defect in fork restart in TFIP11-depleted cells (Supplementary Fig. 4C–E).

It is well established that formation of reversed forks is a prerequisite for MRE11-dependent nascent DNA degradation in BRCA1/2-deficient cells in response to replication stress[17,59,60]. Having shown that TFIP11 is required for efficient replication fork reversal, we next sought to determine whether knockdown of TFIP11 will prevent nascent DNA

**Fig. 3 | TFIP11 accumulates at stalled forks. A** Input and iPOND samples were analyzed by Western blotting. HeLa cells were labeled with 10 μM EdU for 15 min followed by a 1 h chase with 10 μM thymidine (Thd). The cells were then challenged with 4 mM HU for 3 h prior to crosslinking with 1% formaldehyde. No Clk, no-click samples; rxn, reaction. **B** Schematic diagram of the PLA. **C, D** HeLa cells expressing empty vector or HA-Flag-tagged TFIP11 were labeled with 10 μM EdU for 15 min before treatment with 4 mM HU for 3 h. Click chemistry was then used to conjugate biotin to EdU. Representative images of PLA foci (red) shown in (**C**). Scale bar, 10 μm. Quantification of the average number of PLA foci per focus-positive cell or the percentage of cells with PLA foci (**D**). Data represent means ± SD from three independent experiments. From left, $n = 168, 168, 168, 168$ cells. Statistical analysis was calculated with two-tailed, unpaired $t$-test. **E** Strategy for generation of SF-TFIP11 knock-in HeLa cell line. **F** The cell lysates were immunoprecipitated with S protein beads and were probed with anti-Flag or anti-TFIP11 antibody. **G–I** SF-TFIP11 knock-in HeLa cells labeled with 10 μM EdU for 15 min before treatment with 4 mM HU for 3 h. PLA was conducted with anti-Flag and anti-biotin antibodies. Representative images of PLA foci (red) shown in (**G**). Scale bar, 10 μm. Quantification of the average number of PLA foci per focus-positive cell or the percentage of cells with PLA foci (**H**). Data represent means ± SD from three independent experiments. From left, $n = 104, 100, 105, 322, 348, 370$ cells. $P$ values were derived from a one-way ANOVA with Tukey's multiple comparisons test. Knockdown efficiency of TFIP11 was confirmed (**I**). Source data are provided as a Source Data file.

degradation caused by BRCA1/2 inactivation. To this end, we monitored the stability of nascent DNA in cells exposed to sequential pulses of IdU and CldU followed by HU treatment to induce fork stalling (Fig. 5I). In agreement with previous studies, knockdown of BRCA1/2 led to a substantial reduction in CldU/IdU ratio upon HU treatment (Fig. 5I–K). Strikingly, this reduction in the CldU/IdU ratio was almost completely reversed by co-depletion of TFIP11 (Fig. 5I–K), indicating that nascent DNA degradation in BRCA1/2-depleted cells is dependent on TFIP11. In support of this notion, downregulation of TFIP11 significantly reduced the formation of chromosomal abnormalities in BRCA1/2-depleted cells upon HU treatment (Supplementary Fig. 4F–H).

## TFIP11 is required for RAD51 recruitment to stalled forks

The central recombinase RAD51 has been shown to act together with DNA translocases, such as SMARCAL1, ZRANB3, and HLTF, to initiate replication fork reversal[3,7]. To gain mechanistic insight into how TFIP11 contributes to stress-induced fork reversal, we explored whether TFIP11 is important for RAD51 recruitment to stalled forks. As shown in Fig. 6A–D, knockdown of TFIP11 dramatically reduced the association of RAD51 with nascent DNA at stalled forks. By contrast, RPA2 accumulation at stalled forks was not affected by depletion of TFIP11 (Supplementary Fig. 5A–D). In addition, wild-type TFIP11, but not the TFIP11-D3 mutant, was able to alleviate the defect in RAD51 accumulation at stalled forks in TFIP11-depleted cells (Fig. 6A–D). These results suggest that TFIP11 promotes RAD51 recruitment to stalled replication forks in a manner dependent on its ability to bind to DNA.

In addition to its critical role in replication fork reversal, RAD51 has an essential role in DSB repair by HR. Therefore, it would be interesting to test whether TFIP11 may also be required for DSB-induced RAD51 foci formation, a surrogate marker of HR activation. Surprisingly, depletion of TFIP11 did not noticeably affect X-ray-induced RAD51 foci formation (Supplementary Fig. 5E–G). Consistently, HR frequency was comparable in TFIP11-depleted cells and control cells (Supplementary Fig. 5H, I). Taken together, these results suggest that TFIP11 might be particularly important for RAD51-mediated replication fork reversal.

## BLM depletion leads to RAD51 accumulation at stalled forks

The observations that BLM has both pro- and anti-recombinogenic activities[40], and that BLM antagonizes RAD51 function in vivo[61,62], prompted us to test whether BLM might modulate RAD51 activity at stalled forks. Indeed, knockdown of either BLM or RMI1 significantly increased the accumulation of RAD51 at stalled forks (Fig. 7A–D and Supplementary Fig. 6A–D). Conversely, cells overexpressing wild-type BLM exhibited reduction in RAD51 association with nascent DNA after replications stress (Fig. 7E, F and Supplementary Fig. 6E). By contrast, overexpression of the helicase-inactivating mutant BLM-K695A (substitution of lysine 695 with alanine) failed to do so (Fig. 7E, F and Supplementary Fig. 6E), suggesting that BLM limits excessive accumulation of RAD51 at stalled forks in a manner dependent on its helicase activity. To investigate the biological consequences of BLM-

mediated dissociation of RAD51 from stalled forks, we monitored formation of reversed fork in cells overexpressing BLM by EM. As shown in Fig. 7G and Supplementary Fig. 7A, overexpression of wild-type BLM, but not the K695A mutant, resulted in a dramatic decrease in the percentage of HU-induced reversed forks. Consistent with the EM data, analysis of fork progression by DNA fiber assay revealed that forks progressed faster in the presence of HU upon overexpression of wild-type BLM but not its catalytically inactive mutant (Fig. 7H, I). Furthermore, overexpression of wild-type BLM but not its catalytically inactive mutant resulted in marked reduction of BrdU foci (Supplementary Fig. 7B–D), indicating that excessive BLM activity at stalled forks may cause defects in fork reversal and slowing. Notably, knockdown of BLM also resulted in dramatic reduction in the frequency of reversed forks following HU treatment (Fig. 7J, K), which is consistent with a previous study[48]. These results together indicate that either too much or too little activities of BLM or RAD51 at stalled forks could result in impaired replication fork reversal. Importantly, and in line with the observation that TFIP11 and BLM have opposing effects on RAD51 recruitment to stalled forks, simultaneous depletion of TFIP11 and BLM largely suppressed the defects caused by single depletion of TFIP11 or BLM (Fig. 7A–D and Fig. 7J, K,). Similarly, the fork slowing defect induced by TFIP11 or RMI1 inactivation was rescued by co-depletion of TFIP11 and RMI1 (Supplementary Fig. 7E–J).

## TFIP11 depletion results in accumulation of BLM at stalled forks

The aforementioned results prompted us to hypothesize that TFIP11 depletion may affect the accumulation of BLM at stalled forks. Indeed, as shown in Fig. 6D and Fig. 8A–C, knockdown of TFIP11 led to a substantial increase in BLM association with nascent DNA after HU treatment. Likewise, RMI1 also accumulated to supra-physiological levels at stalled forks in TFIP11-depleted cells (Fig. 6D and Supplementary Fig. 8A–C), indicating that TFIP11 may limit the deposition of the BLM complex at stalled forks. Strikingly, the increased association of BLM and RMI1 with nascent DNA in TFIP11-depleted cells could be reversed by re-expression of wild-type TFIP11 but not its GCFC-domain deletion mutant, indicating that the DNA-binding activity of TFIP11 is critical for its function in preventing excessive accumulation of BLM at stalled forks (Figs. 6D, 8A–C and Supplementary Fig. 8A–C). In support of this conclusion, overexpression of wild-type TFIP11 but not its GCFC-domain deletion mutant substantially reduced the recruitment of BLM to stalled forks (Supplementary Fig. 9A–D).

To extend and validate the above findings, we performed DNA pulldown experiments with BLM pre-bound to the splayed-arm DNA followed by the addition of increasing amounts of TFIP11 (Fig. 8D, E). As shown in Fig. 8F, G, pre-bound BLM was displaced by TFIP11 but not by the His-SUMO proteins. Furthermore, in line with this, TFIP11 inhibited the DNA unwinding ability of BLM TFIP11 in vitro (Supplementary Fig. 9E). Importantly, this inhibitory effect relied on TFIP11's ability to bind to DNA, as evidenced by the observation that the TFIP11 mutant lacking its DNA binding activity failed to suppress the binding of BLM with splayed-arm DNA (Fig. 8H and Supplementary Fig. 9F). Conversely, pre-bound TFIP11 can also be displaced by BLM

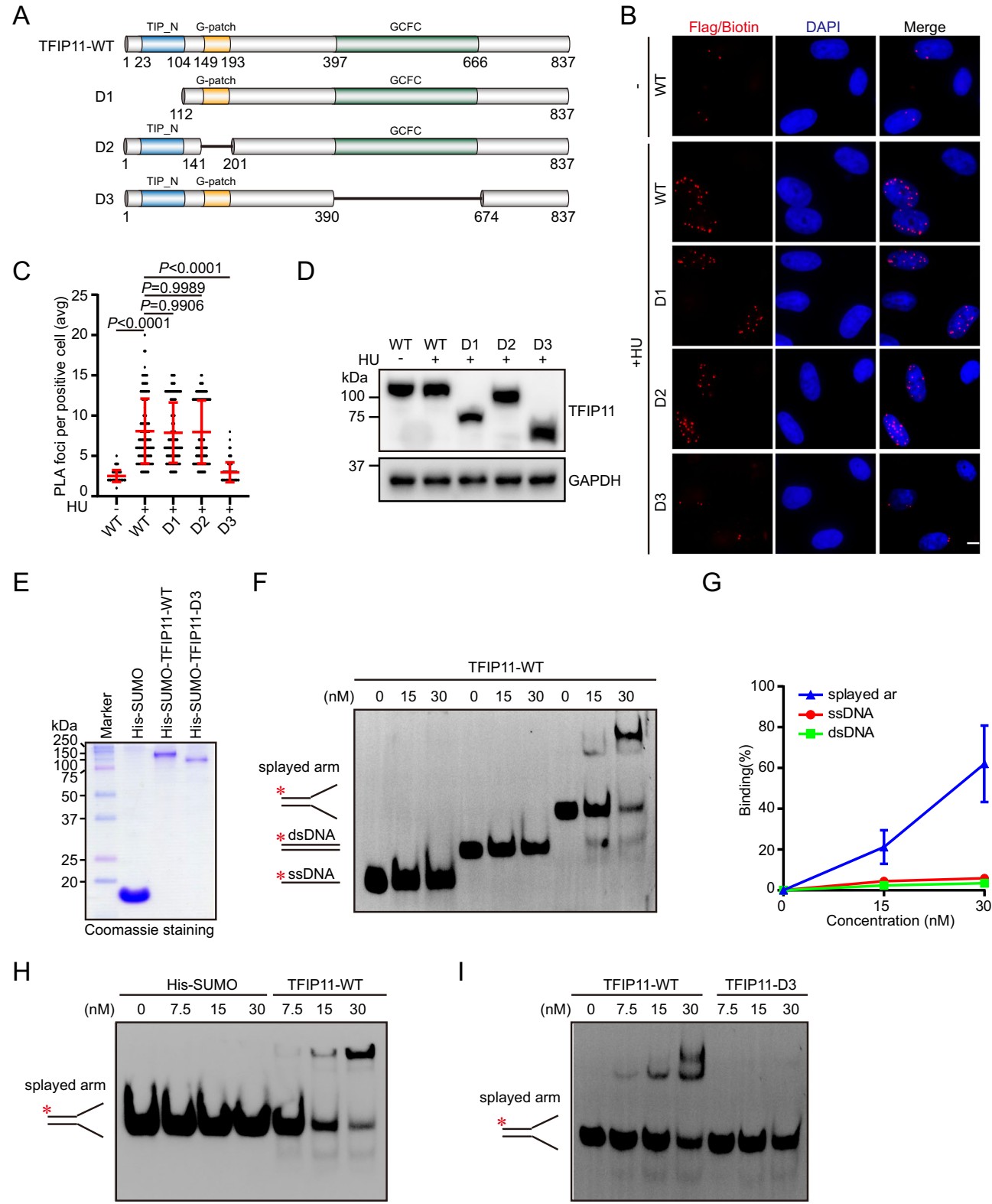

(Supplementary Fig. 9G). Taken together, these results suggest that TFIP11 may modulate BLM and RAD51 activities at stalled forks, thereby promoting replication fork reversal.

## Discussion

RAD51-mediated replication fork reversal has emerged as a global cellular response to a variety of exogenous and endogenous replication stress in higher eukaryotic cells[3]. In this study, we have provided several lines of evidence to show that TFIP11 is a critical regulator of this process. First, TFIP11 exhibits the ability to accumulate at sites of replication stress and rapidly dissociates from stalled forks following stress removal, showing a temporal pattern akin to that of Rad51 and BLM (Supplementary Fig. 10). Second, TFIP11 associates with BLM, and they seem to antagonize each other, contributing to the tight regulation of RAD51 activity at stalled forks. Third, TFIP11 possesses intrinsic DNA-binding activity, a feature that was necessary for its

**Fig. 4 | The GCFC domain of TFIP11 mediates its recruitment to stalled forks.**
**A** Schematic illustration of TFIP11 and its mutants. **B–D** HeLa cells expressing Flag-tagged wild-type TFIP11 or its mutants were labeled with 10 µM EdU for 15 min before treatment with 4 mM HU for 3 h. PLA was conducted with anti-Flag and anti-biotin antibodies. Representative images of PLA foci (red) shown in (**B**). DNA was stained with DAPI. Scale bar, 10 µm. Quantification of the average number of PLA foci per focus-positive cell (**C**). Data represent means ± SD from three independent experiments. From left, $n = 108, 108, 108, 108, 108$ cells. $P$ values were derived from a one-way ANOVA with Tukey's multiple comparisons test. Western blot analysis of TFIP11 expression (**D**). **E** Purified His-SUMO-tagged wild-type and TFIP11 mutants were analyzed by SDS-PAGE followed by Coomassie blue staining. **F** Comparison of the DNA-binding activity of TFIP11 on ssDNA, dsDNA, and splayed-arm substrates.

The reaction mixtures (20 µl) contained increasing amounts of purified His-SUMO-tagged TFIP11 (0, 15, 30 nM) and biotin-labeled DNA substrates (0.5 nM each). **G** Quantification of the results shown in (**F**). Data are the average of three independent experiments and are presented as mean ± SD. **H** TFIP11 but not His-SUMO binds to splayed-arm substrates. The reaction mixtures (20 µl) contained increasing amounts of purified His-SUMO (0, 7.5, 15, 30 nM) or His-SUMO-tagged TFIP11 (7.5, 15, 30 nM) and biotin-labeled DNA substrates (0.5 nM each). **I** The GCFC domain of TFIP11 is required for its DNA-binding activity. The reaction mixtures (20 µl) contained increasing amounts of purified His-SUMO-tagged TFIP11 or its mutant (0, 7.5, 15, 30 nM) and biotin-labeled DNA substrate (0.5 nM). Source data are provided as a Source Data file.

re-localization to stalled forks and its function in promoting fork reversal. Finally, depletion of TFIP11 renders cells more sensitive to agents that induce replication stress and enhances replication stress-induced genome instability. Our findings provide important insights into how the activities of BLM and RAD51 at stalled forks are precisely controlled.

Cells lacking BLM exhibit heightened sensitivity to replication stress-inducing agents and impaired recovery from stalled replication forks, suggesting a compromised response to replication stress. However, the intricate mechanisms through which BLM stabilizes stalled forks remain incompletely understood. An appealing model posits that BLM achieves this by facilitating the regression of a stalled replication fork into a "chicken-foot" structure, thereby averting fork collapse. Indeed, BLM has been demonstrated to catalyze the regression of a model replication fork in vitro[48]. In this study, using a combination of PLA, iPOND, EM, native BrdU staining, and single-molecule DNA fiber approaches, we demonstrate that BLM-dependent fine-tuning of RAD51 activity at stalled forks[40,61,62] is crucial for replication stress-induced fork reversal and slowing in vivo. Notably, similar to BLM depletion, knockdown of RMI1 also causes excessive accumulation of RAD51 at stalled forks and largely rescues the fork slowing defect of TFIP11-inactivated cells, indicating that TFIP11 and the functional BLM complex may antagonize each other at stalled forks to promote RAD51-mediated fork reversal. Strikingly, either overexpression or knockdown of BLM resulted in defects in fork reversal and slowing after HU treatment. Although the reason that depletion of BLM resulted in a reduction in the frequency of reversed forks following HU treatment remains unresolved, at least two possibilities can be envisaged. First, in the absence of BLM, sustained RAD51 at stalled forks may prevent engagement of DNA translocases such as SMARCAL1, ZRANB3 or HLTF, thereby limiting fork reversal. Second, defective removal of RAD51 in BLM-depleted cells may impair its recycling efficiency and, in turn, cause a reduction in the frequency of fork reversal. Thus, the activities of these proteins must be tightly controlled to prevent the otherwise deleterious effects arising from replication stress on cell viability and genome stability.

TFIP11 and BLM are engaged in an interaction where they antagonize each other to promote fork reversal. However, the precise biological significance of this interaction is currently unclear. We speculate that the interaction between TFIP11 and BLM may induce a conformational change in each protein, thereby favoring their release from ssDNA at stalled forks. Thus, TFIP11 and BLM may antagonize each other via two possible mechanisms: (1) by competing for binding sites at stalled forks; (2) by inducing structural alterations in each other. Aside from BLM helicase, several DNA translocases such as SMARCAL1, ZRABB3, HLTF, RECQ1, and FANCM have also been implicated in the replication fork reversal and/or restart process upon replication stress[3–6,9–12]. Investigating the potential interactions and regulatory mechanisms involving TFIP11 and these enzymes represents an interesting avenue for future research.

In addition to TFIP11 and the BLM complex, several proteins are known to play direct or indirect roles in regulating RAD51 activity at stalled forks. For instance, the MMS22L-TONSL protein complex physically interacts witSIRT1h RAD51, facilitating its recruitment to stalled forks[63–68]. Disruption of this complex significantly reduces the frequency of reversed forks induced by replication stress[63–68]. In addition, the ssDNA-binding protein RADX directly interacts with RAD51, counteracting RAD51 accumulation at stalled forks[69,70]. Interestingly, RADX exhibits dual roles in regulating fork reversal, depending on the extent of replication stress[71]. The RAD51 paralog complex BCDX2 (RAD51B-RAD51C-RAD51D-XRCC2) has been shown to assist RAD51 to drive reversed replication fork formation[72]. Notably, the SUMO E3 ligase NSMCE2 prevents excessive RAD51 accumulation at stalled forks[73]. Despite these known interactions, the detailed mechanistic cooperation among these proteins to support optimal RAD51 activities at stalled forks remains unclear and requires further investigation. It is noteworthy to mention that, unlike the MMS22L-TONSL and BCDX2 complexes, TFIP11 depletion did not affect DSB-induced RAD51 foci formation and HR repair. More importantly, inactivation of TFIP11 also had no obvious effect on the frequency of sister chromatid exchange (SCE) (Supplementary Fig. 11). These findings highlight the possibility that TFIP11 and BLM might selectively antagonize each other at stalled forks but not at DSBs or collapsed forks. In accordance with this hypothesis, TFIP11 possesses intrinsic DNA-binding ability with a strong preference for DNA substrates that mimic structure generated at stalled forks. Given that TFIP11-depleted cells are hypersensitive to HU, an agent that induce robust fork reversal, but not to X-ray radiation which causes DSBs, it is tempting to speculate that promotion of RAD51-mediated fork reversal represents the essential activity of TFIP11. TFIP11 is an evolutionarily conserved GCFC domain-containing protein that has recently been described to play an important role in RNA processing[50–53]. In the absence of replication stress, TFIP11 displays a unique and characteristicspeckled nuclear localization pattern, similar to its *C. elegans* homolog STIP (septin and tuftelin-interacting proteins)[74]. However, upon replication stress, it re-localizes to stalled replications forks. Whether this replication stress-induced re-localization of TFIP11 may tempo-spatially couple the replication stress response and RNA processing warrants further investigation.

## Methods

### Cell culture
HEK293T, U2OS, and HeLa cells were cultured in Dulbecco's modified Eagle's medium (DMEM) supplemented with 10% fetal bovine serum (FBS) and 1% penicillin and streptomycin. Prior to use, all cell lines underwent mycoplasma contamination testing. Additionally, cultures were maintained for no longer than one month.

### Antibodies and chemicals
Polyclonal anti-TFIP11 (WB dilution: 1:1000), anti-RAD51 (WB dilution: 1:1000; immunostaining dilution: 1:3000), anti-BLM (WB dilution: 1:1000; immunostaining dilution:1:20 000) and anti-RPA2 (WB dilution:1:1000; immunostaining dilution: 1:20000) were generated by immunizing rabbits with MBP-TFIP11 (amino acids 1–300), MBP-RAD51 full length, MBP-BLM (amino acids 1–449), and MBP-RPA2 full length

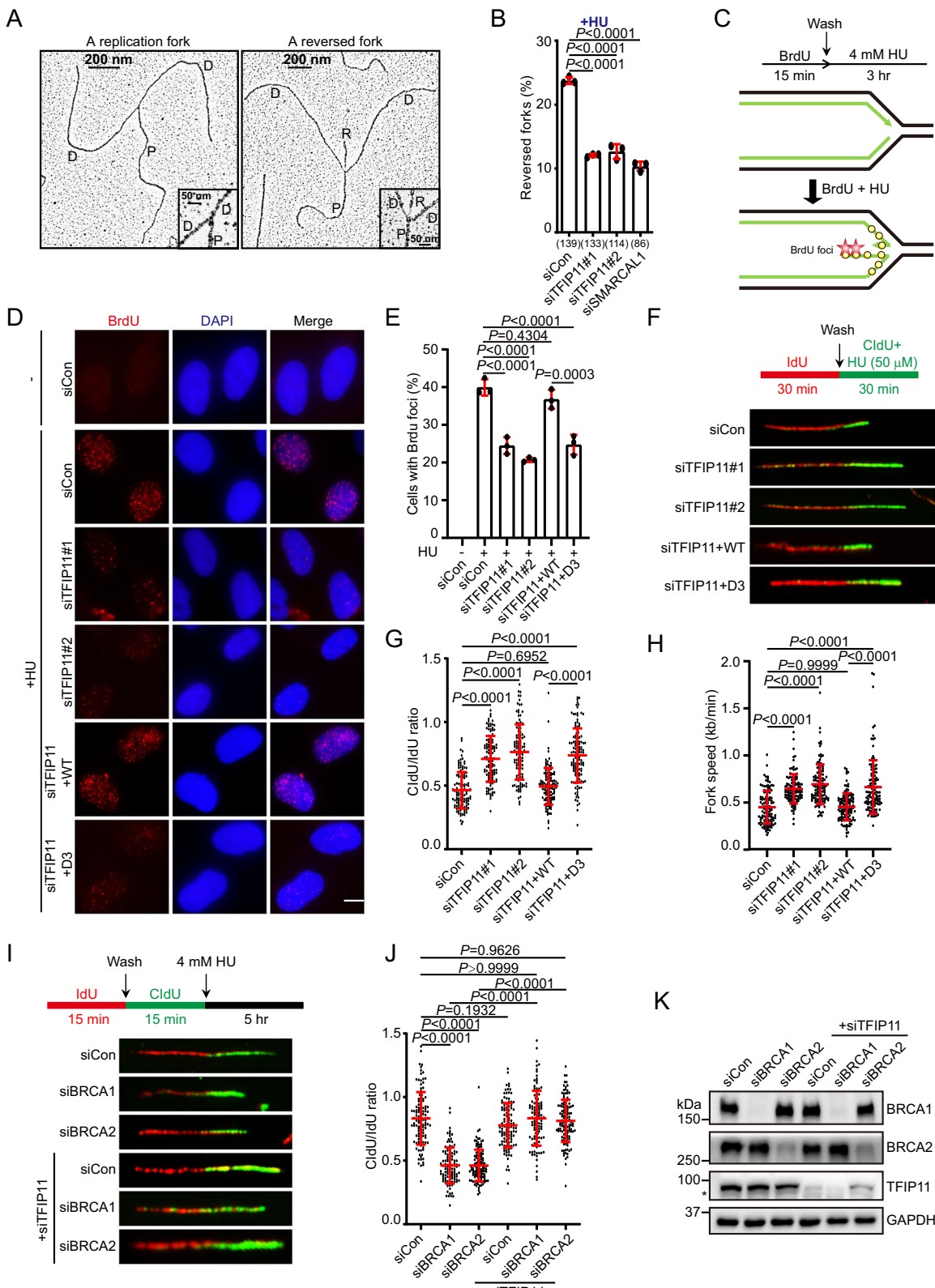

fusion proteins expressed and purified from *E. coli* (Hangzhou HuaAn Biotechnology Co., Ltd). Antisera were affinity-purified using the AminoLink Plus immobilization and purification kit (Thermo Fisher Scientific). Anti-Myc (M20002S, WB dilution: 1:5000) and anti-CtIP (61141, Clone: 14-1, 1:1000 dilution) antibodies were purchased from Abmart and Active Motif, respectively. Anti-CldU/BrdU (ab6326, immunostaining dilution: 1:500) and anti-H3 (04–928, WB dilution:

1:5000) antibodies were purchased from Abcam and EMD Millipore, respectively. Anti-PCNA (PC10) (sc-56, WB dilution: 1:1000) and anti-Flag (M2, WB dilution: 1:5000, immunostaining dilution: 1:1000) antibodies were purchased from Santa Cruz Biotechnology and Sigma-Aldrich, respectively. Anti-BRCA2 (A303-434A, WB dilution: 1:2000), anti-Biotin (150–109 A, immunostaining dilution: 1:3000) and anti-RMI1 (A300-631A, WB dilution:1:1000), and anti-BRCA1 (A301-378A,

**Fig. 5 | TFIP11 promotes fork reversal. A** Electron micrograph illustrating a representative replication fork or reversed fork. P denotes parental duplex, D indicates daughter duplexes, and R indicates the regressed arm. **B** Frequency of reversed forks. Data presented as mean ± SD from three independent experiments, with the total number of analyzed replication intermediates across three replicates shown in brackets. *P* values were derived from a one-way ANOVA with Tukey's multiple comparisons test. **C** Top: schematic of the native BrdU assay. Bottom: Model for the native BrdU assay. Black and green lines represent template and nascent DNA strands, respectively. **D**, **E** HeLa cells were labeled with 10 μM BrdU for 15 min before treatment with 4 mM HU for 3 h. Cells were then fixed and stained with antibody against BrdU without DNA denaturation. Representative BrdU foci shown in (**D**). Scale bar, 10 μm. Quantification of BrdU foci (**E**). Data represent mean ± SD from three independent experiments. From left, *n* = 480, 538, 534, 775, 413, 467 cells. *P* values were derived from a one-way ANOVA with Tukey's multiple comparisons test. **F** Top: schematic representation of the DNA fiber assay. Bottom:

representative fiber images. **G** Dot plot of CldU/IdU ratios for individual replication forks. The results shown represent the mean ± SD of three independent experiments. From left, *n* = 116, 115, 115, 119, 121 cells. *P* values were derived from a one-way ANOVA with Tukey's multiple comparisons test. **H** Fork speed determined by DNA fiber assay. The results shown represent the mean ± SD of three independent experiments. *P* values were derived from a one-way ANOVA with Tukey's multiple comparisons test. **I** Top: schematic representation of the DNA fiber assay. HeLa cells were sequentially labeled with IdU and CldU for 15 min and then challenged with 4 mM HU for 5 h. Bottom: representative fiber images. **J** Dot plot of CldU/IdU ratios for individual replication forks. The results shown represent the mean ± SD of three independent experiments. From left, *n* = 113, 107, 118, 108, 105, 120 fibers. *P* values were derived from a one-way ANOVA with Tukey's multiple comparisons test. **K** Knockdown of TFIP11 and BRCA1/2. Asterisk indicates a non-specific band. Source data are provided as a Source Data file.

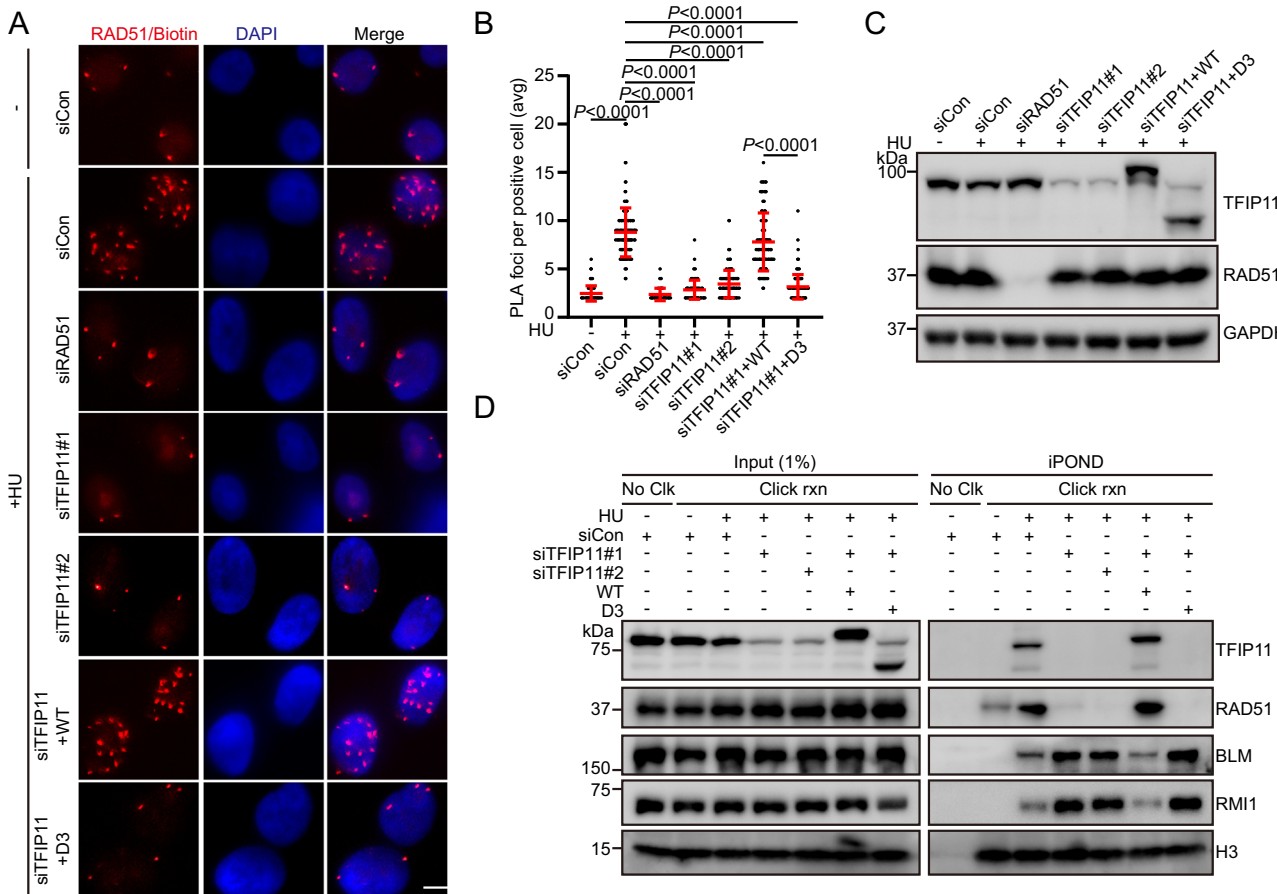

**Fig. 6 | TFIP11 is required for RAD51 recruitment to stalled forks. A−C** TFIP11 is required for RAD51 recruitment. HeLa cells were labeled with 10 μM EdU for 15 min before treatment with 4 mM HU for 3 h. PLA was conducted with anti-RAD51 and anti-biotin antibodies. Representative images of PLA foci (red) shown in (A). DNA was stained with DAPI. Scale bar, 10 μm. Quantification of the average number of PLA foci per focus-positive cell (B). Data represent means ± SD from three independent experiments. From left, *n* = 168, 168, 168, 168, 168, 168, 168 cells. *P* values

were derived from a one-way ANOVA with Tukey's multiple comparisons test. Western blot analysis of TFIP11 expression (**C**). Asterisk indicates a non-specific band. **D** Input and iPOND samples were analyzed by Western blotting. HeLa cells stably expressing wild-type TFIP11 or the D3 mutant were transfected with the indicated siRNAs. 48 h after transfection, cell were labeled with 10 μM EdU for 15 min. Source data are provided as a Source Data file.

WB dilution: 1:1000) antibodies were purchased from Bethyl. Anti-IdU/ BrdU (B44) (347580, immunostaining dilution: 1:500) antibody was purchased from BD Biosciences. Rhodamine conjugated goat anti-mouse IgG (15-001-003, immunostaining dilution: 1:1000) and anti-biotin (200-002-211, immunostaining dilution: 1:3000) antibodies were purchased from Jackson ImmunoResearch. Alexan Fluor 488 Donkey anti-Rat IgG (A-21208, immunostaining dilution: 1:500) was purchased from Life technologies. Anti-BrdU (RPN202,

immunostaining dilution: 1:1000) was purchased from GE. Colcemid (C3915), Hydroxyurea (H8627) and Mitomycin C (HY-13316) were purchased from Sigma-Aldrich.

**Plasmids and transfection**
All cDNAs underwent amplification by PCR and subsequent subcloning into the pDONR201 vector, utilizing the Gateway Technology (Invitrogen). The resulting entry clone was subsequently transferred into a

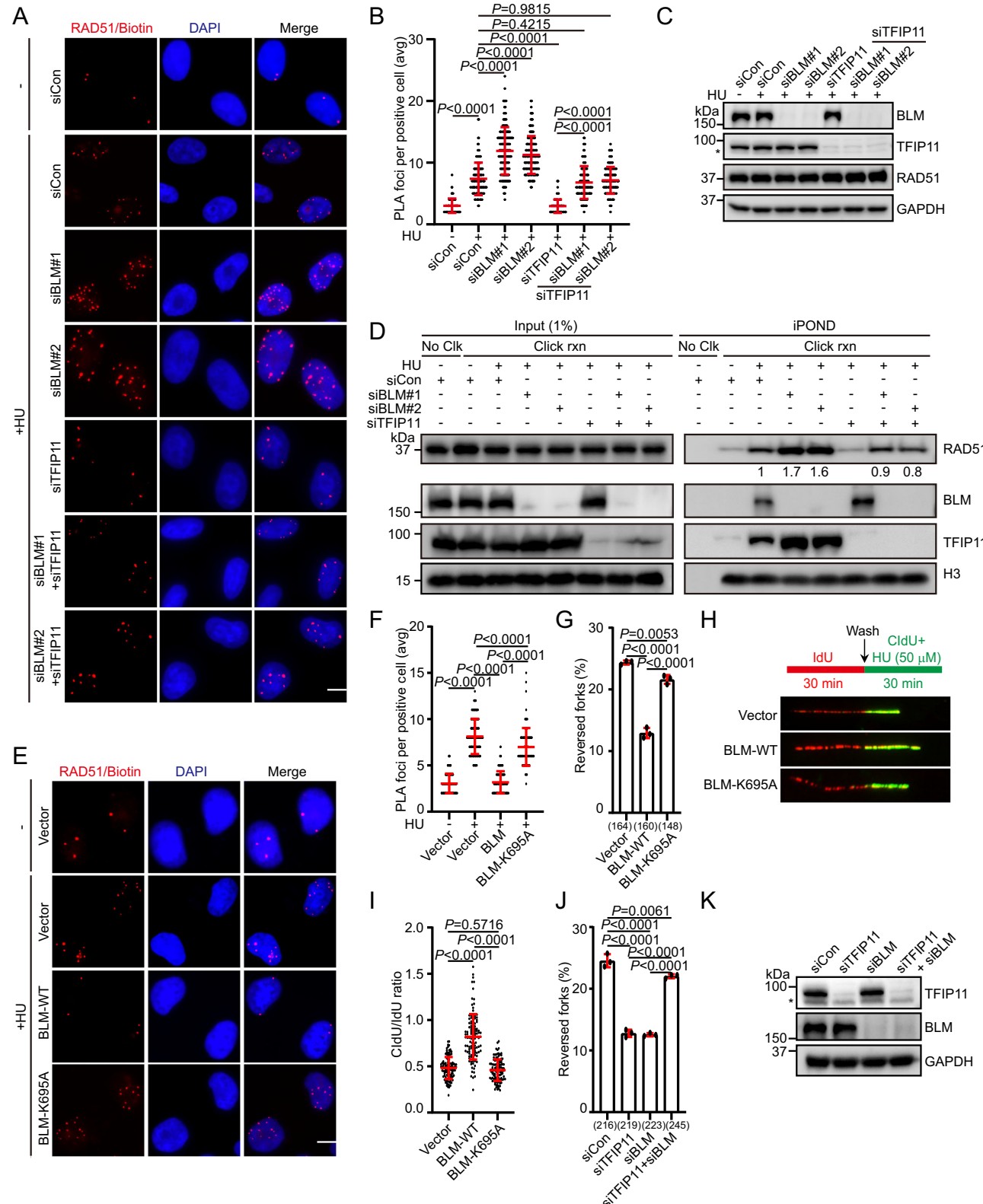

Gateway-compatible destination vector, facilitating the expression of N-terminally SFB-tagged (comprising S tag, Flag epitope tag, and Streptavidin-binding peptide tag), Flag-tagged, or Myc-tagged fusion proteins. Site-directed mutagenesis of TFIP11 and BLM was performed via PCR, following standard procedures, to generate mutants. The fidelity of all constructs was confirmed through DNA sequencing. For

transfections in HEK293T cells, plasmids were introduced using polyethyleneimine (Yeasen Biotechnology) following the manufacturer's instructions. In the case of transfections in HeLa or U2OS cells, plasmids were introduced using Hieff Trans™ Liposomal Transfection Reagent (Yeasen Biotechnology) following the manufacturer's guidelines.

**Fig. 7 | TFIP11 antagonizes BLM to promote RAD51 recruitment.**
**A**–**C** Representative images of PLA foci (red) (**A**). Scale bar, 10 μm. Quantification of the average number of PLA foci per focus-positive cell (**B**). Data represent means ± SD from three independent experiments. From left, $n$ = 145, 144, 144, 150, 144, 144, 144 cells. $P$ values were derived from a one-way ANOVA with Tukey's multiple comparisons test. Knockdown efficiency of TFIP11 and BLM (**C**). Asterisk indicates a non-specific band. **D** Input and iPOND samples were analyzed by Western blotting. Band intensity of RAD51 quantified using ImageJ software and shown below the blot. Representative images of PLA foci (red) (**E**). Scale bar, 10 μm. Quantification of the average number of PLA foci per focus-positive cell (**F**). Data represent means ± SD from three independent experiments. From left, $n$ = 144, 162, 136, 136 cells. $P$ values were derived from a one-way ANOVA with Tukey's multiple comparisons test. **G** Frequency of reversed forks. Data presented as mean ± SD from three independent experiments, with the total number of analyzed replication intermediates across three replicates shown in brackets. $P$ values were derived from a one-way ANOVA with Tukey's multiple comparisons test. **H** Overexpression of BLM compromised fork slowing upon replication stress. Top: schematic representation of the DNA fiber assay. HeLa cells were labeled with IdU for 30 min, and subsequently CldU in the presence of 50 μM HU for 30 min. Bottom: representative fiber images. **I** Dot plot of CldU/IdU ratios for individual replication forks. Data represent means ± SD from three independent experiments. From left, $n$ = 112, 114, 114 fibers. $P$ values were derived from a one-way ANOVA with Tukey's multiple comparisons test. **J** Frequency of reversed forks. Data presented as mean ± SD from three independent experiments, with the total number of analyzed replication intermediates across three replicates shown in brackets. $P$ values were derived from a one-way ANOVA with Tukey's multiple comparisons test. **K** Knockdown efficiency of TFIP11 and BLM. Asterisk indicates a non-specific band. Source data are provided as a Source Data file.

## TAP of SFB-TFIP11 protein complexes and mass spectrometry

HEK293T cells with stable expression of SFB-tagged TFIP11 were lysed with ice-cold NETN buffer (20 mM Tris-HCl [pH 8.0], 100 mM NaCl, 1 mM EDTA, and 0.5% Nonidet P-40) supplemented with 50 U/μl Benzonase and protease inhibitors (1 μg ml$^{-1}$ aprotinin and leupeptin) on ice for 30 min. Crude lysates were cleared by centrifugation at 14,000 × $g$, 4 °C for 8 min, and the resulting supernatant was subjected to incubation with streptavidin-conjugated beads (GE Healthcare) at 4 °C for 2 h with gentle rocking. Following three washes with ice-cold NETN buffer, the immunocomplexes were eluted with biotin (1 mg ml$^{-1}$, Sigma) at 4 °C for 1 h. The elutes were further incubated with S-protein beads (EMD Millipore) at 4 °C for 2 h with gentle rocking. The proteins bound to S-protein beads underwent three washes with ice-cold NETN buffer, were resolved on SDS–PAGE, and then subjected to analysis by mass spectrometry.

## LC-MS/MS analysis

The LC–MS/MS analysis utilized an Easy-nLC system in conjunction with a Thermo Q-Exactive HF mass spectrometer. Desalted peptides were reconstituted in buffer A (0.1% formic acid, 1.9% acetonitrile, 98% H$_2$O) and subjected to analysis on an analytical column. Peptide separation occurred over a 60-min linear gradient at a flow rate of 600 nl/min, with the following gradient steps: 6–15% buffer B (0.1% formic acid, 99.9% acetonitrile) for 16 min, 15–26% buffer B for 35 min, 26–42% buffer B for 15 min, 42–95% buffer B for 1 min, and 95% buffer B for 8 min. The mass spectrometer employed data-dependent mode for peptide detection, initiating with a full MS scan at R = 120,000 (m/z 200). This was followed by twenty HCD MS/MS scans at R = 15,000, utilizing a normalized collision energy (NCE) of 27 and an isolation width of 1.6 m/z. AGC targets for MS1 scans were set at $3 \times 10^6$, while for MS2 scans, the AGC targets were $2 \times 10^4$. The maximum injection time for MS1 was 80 ms, whereas for MS2, it was 20 ms. Dynamic exclusion was implemented for 12 s, excluding precursors with unassigned charge states or charge states of 1+, >6+. Mass spectrometry data were analyzed using MaxQuant (version 2.3), and the human protein database was sourced from NCBI (https://www.ncbi.nlm.nih.gov/). Precursor ions and fragment ions had tolerances of 10 ppm and 20 ppm, respectively, with a maximum of 2 allowed miscleavage sites. The protein interactome list generated is provided in Supplementary Data 1.

## Coimmunoprecipitation and Western blotting

HeLa cells were either untreated or exposed to 4 mM HU for 3 h. Following treatment, cells were lysed using NETN buffer supplemented with 50 U/μl Benzonase and protease inhibitors on ice for 20 min. Following sonication, the cell lysates were centrifuged at 14,000 × g at 4 °C for 5 min, and the resulting supernatants were incubated with 25 μl protein A–Sepharose coupled with 2 μg of the specified antibodies at 4 °C for 3 h. Beads were then subjected to three washes with NETN buffer, eluted with 2 × SDS loading buffer, and resolved on SDS–PAGE. Immunoblotting was carried out according to standard procedures.

## siRNA and transfection

For siRNA experiments, cells were seeded in a 6-well plate and transfected with siRNAs using Lipofectamine RNAiMAX reagent (Invitrogen) according to the manufacturer's instructions. siRNAs were purchased from Ruibo, and the siRNA sequences were as follows: siCon: 5'-UUCAAUAAAUUCUUGAGGUUU-3'; TFIP11 siRNA#1: 5'-GCAC AAGGUAUCAUUAACC dTdT-3'; and TFIP11 siRNA#2: 5'-GCACAA CGUUCCCGAUGAUdTdT-3'; BLM siRNA#1: 5'-GUACUAAAUGGCAAU UUAAdTdT-3'; and BLM siRNA#2: 5'-GCUAGGAGUCUGCGUGC GAd TdT-3'; RMI1 siRNA#1: 5'-AGCCUUCACGAAUGUUGAUdTdT-3'; and RMI2 siRNA#2: 5'-UCUAGUUACAGCUGAAGCAdTdT-3'; BRCA2 siRNA: 5'-GGGAAACACUCAGAUAAA dTdT-3'; BRCA1 siRNA: 5'-AGAUAGUUCU ACCAGUAAAdTdT-3'; CtIP siRNA: 5'-GCUAA AACAGGAACGAA UCdTdT-3'. The siRNA-resistant wild-type and mutant TFIP11 plasmids were constructed by substituting seven nucleotides in the TFIP11 siRNA#1-targeting region (A516G, A519G, T522A, C525A, T528C, C531T and A534G).

## iPOND assay

For the iPOND assay, logarithmically growing HEK293T cells were subjected to the following procedure: initially, cells were treated with 10 μM EdU for 15 min, followed by a 1-h thymidine (10 μM) chase. Subsequently, cells were exposed to 4 mM HU for 3 h and then fixed with 1% formaldehyde for 20 min. After fixation, cells were harvested, quenched with 0.125 M glycine, and washed with PBS. The resulting cell pellets were permeabilized with 0.25% Triton X-100 for 30 min, followed by a wash with ice-cold 0.5% BSA/PBS and another wash with PBS. Next, the cells were incubated with click reaction buffer (10 mM sodium ascorbate, 2 mM CuSO4, and 10 μM biotin azide) for 80 min. After the Click reaction, cells were washed once with ice-cold 0.5% BSA/PBS and once with PBS. Subsequently, cells were resuspended in lysis buffer (1% SDS, 50 mM Tris [pH 8.0], 1 μg/ml aprotinin and leupeptin) and subjected to sonication. Lysates were then cleared and incubated with Streptavidin-agarose beads at 4 °C overnight. The beads underwent washes, including once with lysis buffer, once with 1 M NaCl, and twice with lysis buffer, before being eluted with 2 × SDS Laemmli buffer for 25 min at 95 °C. The resolved proteins were separated by SDS-PAGE and subsequently immunoblotted with the indicated antibodies.

## Proximity ligation assay

HeLa cells were seeded onto coverslips in a 6-well plate and allowed to incubate for 24 h. Following this, cells were labeled with 10 μM EdU for 15 min followed by exposure to 4 mM HU for 3 h. After PBS washing, the cells were permeabilized with 0.5% Triton X-100 at 4 °C for 10 min, fixed with a solution of 3% formaldehyde and 2% sucrose for 10 min, and blocked with 3% BSA for 30 min. After three PBS

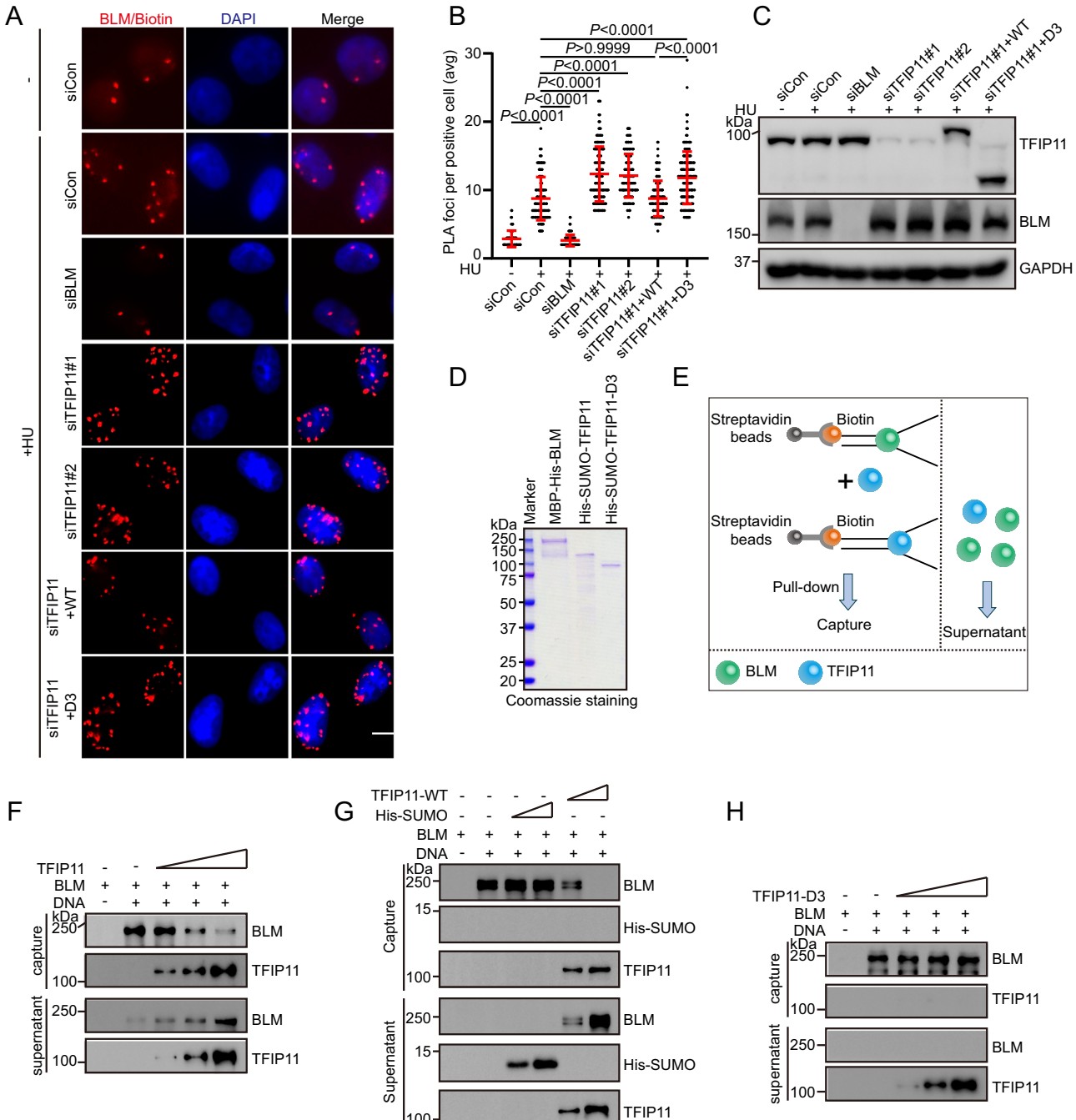

**Fig. 8 | TFIP11 modulates BLM interaction at stalled forks. A–C** HeLa cells were labeled with 10 µM EdU for 15 min before treatment with 4 mM HU for 3 h. PLA was conducted with anti-BLM and anti-biotin antibodies. Representative images of PLA foci (red) (**A**). Scale bar, 10 µm. Quantification of the average number of PLA foci per focus-positive cell (**B**). Data represent means ± SD from three independent experiments. From left, n = 168, 168, 168, 168, 168, 168, 168 cells. *P* values were derived from a one-way ANOVA with Tukey's multiple comparisons test. Western blot analysis of TFIP11 expression (**C**). Asterisk indicates a non-specific band. **D** SDS-PAGE profile of purified MBP-His-tagged BLM and His-SUMO-tagged TFIP11. **E** Schematic representation of the competition assay. **F, H** BLM (24 nM) was added to biotinylated splayed-arm DNA (100 nM) coupled to magnetic streptavidin beads

in 20 µl of binding buffer (20 mM Tris-HCl, PH 7.5, 120 mM NaCl, 0.1% Triton X-100, 2 mM CaCl$_2$, 10 mM Mg(OAc)$_2$, 1 mM DTT, 0.1 mg/mL BSA) at 4 °C for 30 min. Excess BLM removed by magnetic separation. Then, increasing amounts of TFIP11 (**F**) or its mutant (12 nM, 24 nM, 48 nM) (**H**) was added and reactions were incubated at 4 °C for 30 min prior to separation of the DNA-bound and supernatant fractions. **G** BLM (24 nM) was added to biotinylated splayed-arm DNA (100 nM) coupled to magnetic streptavidin beads in 20 µl of binding buffer at 4 °C for 30 min. Excess BLM removed by magnetic separation. Then, increasing amounts of His-SUMO (24 nM, 48 nM) or TFIP11 (24 nM, 48 nM) was added and reactions were incubated at 4 °C for 30 min prior to separation of the DNA-bound and supernatant fractions. Source data are provided as a Source Data file.

washes, the cells were subjected to a "Click-iT reaction". Following that, cells were incubated with primary antibodies at 4 °C overnight. The proximity ligation assay was performed using the Duolink In Situ Red Starter kit (Sigma-Aldrich) following the manufacturer's

instructions. Images were captured using a fluorescence Microscope (Eclipse 80i; Nikon) equipped with a Plan Fluor 60 × oil objective lens (NA 0.5–1.25; Nikon) and a camera (CoolSNAP HQ2; Photometrics).

## DNA fiber analysis

To assess replication fork restart, HeLa cells were exponentially growing and labeled with 50 μM IdU for 15 min before treatment with 4 mM HU for 3 h. Following HU treatment, cells were labeled with 50 μM CldU for an additional 15 min. For monitoring replication fork progression, exponentially growing HeLa cells were labeled with 50 μM IdU for 30 min, and subsequently with 50 μM CldU in the presence of 50 μM HU for another 30 min. To investigate nascent strand degradation, exponentially growing HeLa cells were sequentially labeled with 50 μM IdU and 50 μM CldU for 15 min each. After the labeling, cells were treated with 4 mM HU for 5 h. Following the respective treatments, cells were collected and lysed with lysis buffer (0.5% SDS in 200 mM Tris-HCl [pH 7.4], 50 mM EDTA). DNA fibers were then spread onto slides, air-dried, fixed in ice-cold methanol/acetic acid (3:1), and denatured with 3 M HCl. After washing with PBS, slides were incubated with anti-IdU/BrdU and anti-CldU/BrdU antibodies. Following a 3 h incubation, slides were incubated with Rhodamine-conjugated goat anti-mouse IgG and Alexan Fluor 488 Donkey anti-Rat IgG for 3 h. DNA fibers were imaged with the Nikon Eclipse 80i Fluorescence Microscope equipped with a Plan Fluor 60 × oil objective lens (NA 0.5–1.25; Nikon) and a camera (CoolSNAP HQ2; Photometrics). More than 100 fibers were measured for each individual experiment.

## Electron microscopy

Electron Microscopy (EM) analysis of replication intermediates was conducted following a previously described protocol with minor modifications[75]. In brief, HeLa cells were exposed to 4 mM HU for 3 h. Following treatment, cells were collected, suspended in ice-cold PBS, and cross-linked with TMP (4,5′,8-trimethylpsoralen, Sigma-Aldrich) for 5 min in the dark. Subsequently, the cells were exposed to UV 365 nm monochromatic light (BLE-7600B, Spectroline) for 3 min. This cycle of TMP treatment, dark incubation, and UV irradiation was repeated five times. The treated cells were lysed with a lysis buffer (1.28 M sucrose, 20 mM MgCl$_2$, 40 mM Tris-HCl [pH 7.5], and 4% Triton X-100) at 4 °C for 10 min and digested with digestion buffer (30 mM EDTA [pH 8.0], 800 mM guanidine–HCl, 5% Tween-20, 30 mM Tris–HCl [pH 8.0], and 0.5% Triton X-100) at 50 °C for 2 h in presence of proteinase K (0.8 mg/ml). Genomic DNA was then purified using chloroform/isoamylalcohol (24:1) and precipitated in one volume of isopropanol. The extracted genomic DNA was washed with 70% ethanol and resuspended in 200 μl TE buffer. A total of 100 U of PvuII HF restriction enzyme was used to digest 15 μg of genomic DNA at 37 °C for 3 h. The digested DNA was loaded onto a pre-equilibrated QIAGEN-tip 20 column with QBT buffer (750 mM NaCl, 50 mM MOPS, 15% isopropanol, 0.15% Triton X-100 [pH 7.0]). Subsequently, the column underwent a washing step with washing buffer (10 mM Tris-HCl [pH 8.0] and 1 M NaCl), followed by equilibration with equilibration buffer (10 mM Tris-HCl [pH 8.0] and 300 mM NaCl). The column was then washed twice with a high NaCl solution (10 mM Tris-HCl [pH 8.0] and 850 mM NaCl), and the elution of DNA was performed in a caffeine solution (10 mM Tris-HCl [pH 8.0], 1 M NaCl, and 1.8% [w/v] caffeine). Finally, eluted DNA were spread on the water surface using the BAC method. The samples were loaded on carbon-coated 400-mesh copper grids and coated with platinum using a High Vacuum Evaporator (MED 020, Leica). Subsequent imaging was carried out using an HT7700 transmission electron microscope equipped with a GATAN camera, and image acquisition was controlled by Digital Micrograph software.

## Cell survival assay

U2OS cells transfected with the indicated siRNAs were plated into 6-well plate in triplicates. At 18 h after plating, cells were either mock-treated or exposed to varying doses of HU or X-ray as indicated. 24 h later, cells were washed with PBS and released into fresh medium for 14 days to facilitate colony formation. The resulting colonies were visualized through Coomassie blue staining and subsequently counted.

## Detection of nascent ssDNA by native BrdU assay

HeLa cells, cultured on coverslips, underwent a 15-min pulse-labeling with 10 μM BrdU (Sigma-Aldrich) before exposure to 4 mM HU for 3 h. After washing with PBS, cells were permeabilized with 0.5% Triton X-100 for 5 min at 4 °C and fixed with 3% paraformaldehyde for 10 min at room temperature. After three PBS washes, the fixed cells were incubated sequentially with anti-BrdU antibody for 30 min and rhodamine-conjugated goat anti-mouse IgG for 30 min. DAPI was used to counterstain the nuclear DNA, and images were captured with a fluorescence Microscope (Eclipse 80i; Nikon) equipped with a Plan Fluor 60 × oil objective lens (NA 0.5–1.25; Nikon) and a camera (Cool-SNAP HQ2; Photometrics).

## Detection of chromosomal aberrations

HeLa cells, transfected with the indicated siRNAs, underwent mock treatment or were exposed to 4 mM HU for 5 h. Following treatment, cells were released into fresh medium containing 200 ng/ml nocodazole for 16 h and then exposed to colcemid (1 μg/mL) for an additional 4 h. Cells were subsequently harvested, subjected to swelling with a hypotonic solution (75 mM KCl) at 37 °C for 15 min, and fixed with a 3:1 methanol:acetic acid solution for 20 min. The cell suspension was dropped onto ice-cold wet glass slides and air-dried. Finally, slides were stained with a 5% Giemsa solution for 5 min and examined by light microscopy. A minimum of 100 metaphase chromosomes were scored from each sample.

## Electrophoretic mobility shift assay (EMSA)

EMSA assays were carried out using the Pierce Lightshift chemiluminescent EMSA kit following the manufacturer's instructions. Briefly, reaction mixtures (20 μl) comprised 10 mM MgCl$_2$, 2.5% glycerol, 0.05% Nonidet P-40, and 0.5 nM biotin-labeled DNA substrates. These mixtures were incubated with the indicated amounts of TFIP11 for 20 min at room temperature. The reactions were terminated by adding 5 μl of loading buffer, resolved by native polyacrylamide gel, and transferred onto PVDF membranes on ice. After cross-linking by UV irradiation (120 mJ per cm$^2$), biotin-labeled DNA was detected by the Chemiluminescent method. All DNA substrates shared a common 5′-biotin-labeled oligo 1 (5′-GACGCTGCCGAATTCTACCAGTGCCTTGC-TAGGAC ATCTTT GCCCACCTGCAGGTTCAC-3′). dsDNA was generated by annealing oligo 1 with oligo 2 (5′-GTGAACCTGCAGGTGG GCAAAGATGTCCTAGCAAGGCACTGGTAGAATT CGGCAGCGTC-3′). The splayed arm substrate was produced by annealing oligo 1 with oligo 3 (5′-ATAGTCGGATCCTCTAGACAGCTCCATGTAGCAAGGCA CTGGTAGAATTCGGCA GCGTC-3′). Substrates were annealed by heating to 95 °C, followed by slow cooling. All DNA substrates were purified by gel electrophoresis.

## BrdU incorporation assays

U2OS cells were subjected to transfection with the specified siRNAs. After 48 h of transfection, BrdU (100 μM) were introduced into the medium for a duration of 1 h. Subsequently, cells were harvested and fixed with ice-cold 70% ethanol. DNA denaturation was achieved by treating with 2.5 M HCl for 1 h at room temperature. Following PBS washing, cells were incubated with anti-BrdU antibody in blocking buffer (PBS + 0.1% Triton X-100 + 5% BSA) for 12 h, followed by washing with blocking buffer containing 500 mM NaCl. FITC conjugated goat anti-mouse IgG was added and incubated for 4 h. Cells were then resuspended in PBS containing propidium iodide (20 μg/mL) and RNase A (200 μg/mL) at 37 °C for 20 min. The cell cycle distribution was analyzed using a FACScan flow cytometer.

## Recombinant protein expression and purification

The coding sequences of both wild-type and mutant TFIP11 were subcloned into the pET28-N-His-SUMO vector (EMD Millipore) and transformed into BL21 *E. Coli* for the expression of His-SUMO-tagged fusion protein. When the OD600 reached 0.6, cells were induced with 0.2 mM IPTG at 16 °C for 16 h. Subsequently, cells were harvested, suspended in lysis buffer (20 mM Tris [PH 8.0], 300 mM NaCl, 1% Triton X-100, 1 mM DTT, and 1 μg ml$^{-1}$ each of leupeptin and aprotinin). After sonication, the extracts were cleared by centrifugation at $40,000 \times g$ for 40 min at 4 °C. The supernatant was collected and subjected to incubation with cobalt agarose for 6 h at 4 °C. Following beads washing with washing buffer I (20 mM Tris [PH 8.0], 500 mM NaCl, 0.5% NP-40, 20 mM imidazole,1 mM DTT, and 1 μg ml$^{-1}$ each of leupeptin and aprotinin), bound proteins were eluted with washing buffer containing 200 mM imidazole. The eluted proteins were loaded on pre-equilibrated Superdex 75 Increase 10/300 GL column (GE Healthcare) and eluted with elution buffer (20 mM Hepes (K$^+$) [PH 7.6], 100 mM KCl, 0.01%(v/v) NP-40, 1.5 mM MgCl$_2$, 0.2 mM EDTA, 20% (v/v) glycerol, 1 mM DTT, 0.2 mM PMSF and 1 μg ml$^{-1}$ each of leupeptin and aprotinin). Peak protein fractions were pooled and analyzed on 10% SDS-PAGE. Full-length BLM were cloned into the pMAL-c2PS-His vector (kindly provided by Dr. Weibin Wang) and transformed into Rosetta *E. coli* cells for the expression of MBP-His-tagged BLM. Cells were grown at 37 °C until the log phase and then induced with 0.2 mM IPTG at 16 °C for 16 h. Following cell harvesting, suspension in lysis buffer, and sonication, the extract was centrifuged at $40,000 \times g$ for 40 min at 4 °C. The supernatant was collected and incubated with Amylose resin for 4 h at 4 °C. After washing the beads with washing buffer II (20 mM Tris [PH 8.0], 500 mM NaCl, 0.5% NP-40, 1 mM DTT, and 1 μg ml$^{-1}$ each of leupeptin and aprotinin), bound proteins were eluted with washing buffer I containing 10 mM Maltose. The eluted protein was then incubated with cobalt agarose for 6 h at 4 °C. The beads were washed three times with washing buffer II containing 20 mM imidazole and twice with elution buffer containing 20 mM imidazole. The bound proteins were then eluted with elution buffer containing 200 mM imidazole on ice, flash-frozen, and stored at −80 °C.

## Analysis of sister chromatid exchanges

HeLa cells, post-transfection with the indicated siRNAs, were cultured in the presence of 100 μM BrdU for 42 h. Subsequently, cells were subjected to treatment with colcemid (0.2 μg/ml, Sigma) 4 h prior to harvesting through mitotic shake-off. After collection, cells were swollen in pre-warmed (37 °C) hypotonic solution (46.5 mM KCl, 8.5 mM NaCitrate) for 15 min, and fixed in ice-cold methanol/acetic acid (3:1). Metaphase cells was then spread onto glass slides and left to air-dry. After a 24 h interval, the slides were subjected to heating at 88 °C for 10 min in buffer (1.0 M NaH$_2$PO$_4$ [pH 8.0]), followed by rinsing in distilled water, staining with 5% Giemsa for 10 min, and a final rinse with water before allowing to air-dry.

## Statistics and reproducibility

The number of samples for each experiment is indicated in the figure legends. For Western blot or EMSA assays, typically, three independent experiments were carried out. All statistical analyses were performed using GraphPad Prism version 8.2.1. $P$ values were derived from a one-way ANOVA with Tukey's multiple comparisons test. Significance was set at $P \leq 0.05$ for all experiments. The number of replicates, statistical tests applied and $P$-values for each analysis are included in the figure legends.

## Reporting summary

Further information on research design is available in the Nature Portfolio Reporting Summary linked to this article.

## Data availability

All data generated or analyzed during this study are included in this article and its Supplementary information. The Mass spectrometry data have been deposited to the ProteomeXchange Consortium via the PRIDE partner repository under accession number PXD042222. Source data are provided with this paper.

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

## Acknowledgements

We thank all members of the Liu group for insightful discussions. This work was supported by National Key Research and Development Program of China (2022YFA1302800 and 2021YFA1101000 to T.L. and J.H.), Natural Science Foundation of Zhejiang Province (LZ24C070002 to J.H.), National Natural Science Foundation of China (31961160725, 31730021, and 31971220 to J.H., 32270769, 31970664, and 31822031 to T.L.), Zhejiang Science Foundation for Distinguished Young Scholars (LR18C070001 to T.L.), Fok Ying Tung Education Foundation, and Fundamental Research Funds for the Zhejiang Provincial Universities (2021XZZX039).

## Author contributions

J.L.C., M.J.W., Y.L.Y., C.Y.R., Y.L., L.Z.S. and T.W. performed the experiments; J.H. and B.Y. participated in experimental design and data analysis. T.L. designed and supervised the project and wrote the manuscript.

## Competing interests

The authors declare no competing interests.
