## [Peer Review File · Nature Communications]

TFIP11 collaborates with the Bloom syndrome helicase to modulate RAD51 activity at stalled replication forks and promote fork reversalREVIEWER COMMENTS

Reviewer #1 (Remarks to the Author):

This manuscript from Chen et al. describes the identification of TFIP11, a splicing factor, as a binding partner of the Bloom's syndrome complex using co-immunoprecipitation techniques. Subsequently, the authors proceed to demonstrate that the depletion of TFIP11 via siRNAs leads to an increased susceptibility to agents inducing replication stress. Moreover, they observe notable modifications in the dynamics of replication forks and an elevated occurrence of chromosomal abnormalities. By employing iPOND and proximity ligation assays (PLAs), the authors assert that TFIP11 is recruited to replication forks upon exposure to hydroxyurea, potentially through direct DNA binding facilitated by its GCFC domain. Furthermore, they highlight the significance of this recruitment in TFIP11's role in promoting fork reversal. The authors end the manuscript by proposing that TFIP11 competes with BLM for access to stalled forks, thereby facilitating RAD51-mediated fork reversal.

While the first part of the authors' manuscript about a role for TFIP11 in the replication stress response is largely convincing, the proposed mechanism involving BLM cannot be correct, as I explain below.

Specific points:

1. The proposed hypothesis by the authors suggests a competitive relationship between TFIP11 and BLM regarding access to stalled forks. The authors suggest that TFIP11, by effectively out-competing BLM, hinders the removal of RAD51, subsequently promoting RAD51-dependent fork reversal. However, several issues arise in relation to this model. Firstly, according to the model, depletion of BLM should result in an increased occurrence of fork reversal. However, contrary to this expectation, the authors actually observe a reduction in the number of reversed forks in BLM-depleted cells, as depicted in Figure 7J. This observed outcome critically undermines the validity of the authors' proposed model.

2. Secondly, according to the authors' proposed model, the loss of BLM should lead to an increased accumulation of RAD51 at stalled forks. However, previous studies, such as Bugreev et al. (JBC 2009) and Xue et al. (NAR 2019), have demonstrated that BLM is incapable of removing the active ATP-bound form of RAD51 from DNA. Additionally, studies by Patel et al. (JCB 2017), Shorrocks et al. (Nat. Comm. 2021), and others, have shown that the loss of BLM does not result in elevated RAD51 foci. It is likely that RECQL5, another RecQ family helicase, plays a much more significant role in countering RAD51 filament formation in human cells, as indicated by Hu et al. (Genes Dev. 2007) and Schwendener et al. (JBC 2010). The iPOND experiment presented in Figure 7D, which the authors claim demonstrates a 1.5-fold increase in RAD51 on nascent DNA in siBLM-treated cells, is not convincing. The observed increase is minimal, and the H3 loading control is over-exposed to the extent that assessing protein loading becomes impossible. Additionally, a similar experiment in Figure S6D involving overexpression of wild-type and helicase-dead BLM is depicted, but the level of BLM overexpression is so excessively high that its biological relevance becomes questionable when considering the extent of helicase overexpression.

3. Thirdly, if the authors' model is correct, then why would BLM and TFIP11 need to interact at all if they compete for the same substrate? BLM is a common contaminant in proteomics experiments (see crapome.org), and TFIP11 seems to interact with thousands of cellular proteins. Can the authors really be sure the BLM-TFIP11 interaction is biologically relevant?

4. Major conclusions in the manuscript heavily rely on a substantial number of PLA experiments. However, I find it perplexing that the authors have opted for this assay to demonstrate protein colocalization, considering its susceptibility to artifacts (as highlighted <https://www.biorxiv.org/content/10.1101/411355v2> for example). The fact that so few PLA foci are

present in cells when there would be hundreds of stalled forks in a cell treated with hydroxyurea is problematic (e.g. Figure 4B). The authors should re-analyze by conventional IF microscopy the recruitment of proteins instead of using PLA.

5. All western blots require molecular weight markers.

Reviewer #2 (Remarks to the Author):

Chen and colleagues' manuscript is a well-written characterisation of a novel genome stability factor TFIP11. The overall conclusions are that TFIP11 promotes replication fork regression by preventing the dissociation of RAD51 from stalled forks by BLM helicase. The data as presented support the hypothesis:

The findings are very interesting, although some weaknesses include:

1. an over-reliance on proximity ligation assay experiments in support of conclusions, many times without the appropriate controls required for accurate reporting of PLA results (see for example <https://www.biorxiv.org/content/10.1101/411355v2.full>). The authors should indicate specifically which controls were used in all PLA experiments, and show that the antibodies that they are using in their experiments do indeed measure colocalization of proteins or protein/DNA, and not just changes in the levels of expression (eg siRNA, overexpression) of one partner relative to the other.

2. the use of bar graphs to present data that is likely hiding what should have a large spread of values. All the experiments measuring PLA numbers/nucleus, stalled forks and EM reversed forks should include the primary data points used to derive the mean and stdev. These graphs have such tight error bars, it is hard to know what "real values" were used to derive the results. The graphs should be redrawn showing the primary data points (similar style to 5H, I, J).

2a) One problem for presenting PLA foci "per positive cell" is that cells without PLA signals are not counted. Does the fraction of PLA positive/negative cells also change, or is it consistent (this is shown for 3D and 3H, but not other PLA graphs)? Showing a value for every nucleus (including those that are negative) is a much more open way of showing the data.

2b) how is the standard deviation derived from the graphs Fig 5B, 5C, 7G, 7J? A number is given below each bar indicating the number of forks counted, but is this the total number of counts across three replicates, or the total number in each replicate?

3. Several experiments use siRNA+transient transfection experiments to show rescue of very sensitive results. The results as presented don't show what *percentage* of cells are transfected in the experiments. Complete rescue of the phenotype would require 100% of cells to be transfected, and expression to be relatively uniform across those cells. But I am skeptical that PEI can achieve 100% or even 50% in U2OS or HeLa cells where most experiments were done. But if this is possible it would be good to see the results, eg shown by FACS of a transfection reporter.

TFIP11 is previously characterised for its role in splicing. Were splicing factors identified in the mass spectrometry experiment of Figure 1? Or were only the indicated DNA repair factors the only partner proteins identified? All interacting proteins should be indicated. The mass spec data should be deposited in the PRIDE repository so that it can be analysed by others.

The authors conclude that TFIP11 blocks BLM activity on forks, and show that it competitively binds with BLM in a DNA pulldown experiments. This is a nice result, but one that could be more significant if an activity (helicase/unwinding) assay was performed with the proteins. This would conclusively demonstrate that TFIP11 is blocking the helicase activity of BLM.

There are many other translocase enzymes that can regress replication and/or restart forks (HLTF, SMARCAL1, ZRANB3, RECQ1, FANCM, etc). There should be some discussion about whether the authors think these enzymes could also be inhibited or promoted by binding of TFIP11 at the junction.

One minor issue to discuss: the authors show that TFIP11 knockdown leads to increased ssDNA (regressed forks), but they see no change in RPA2 foci formation. These results seem to conflict as most (all?) ssDNA is usually RPA coated in cells. Please comment on whether the ssDNA at regressed forks is bound by something other than RPA.

The paper is well written and the function of TFIP11 described is novel. There is a lot of work done within the paper that supports the conclusions. Addressing the mostly data-presentation related issues above will improve the robustness and reproducibility of the manuscript.

Reviewer #3 (Remarks to the Author):

Manuscript ID: NCOMMS-23-18765

Title: TFIP11 competes with the Bloom syndrome helicase for binding to stalled replication forks to preserve genome stability

Authors: Chen et al.,

In this manuscript the authors have tried to determine the role of GCFC domain-containing protein TFIP11 (also known as TIP39) after replication stress. Using a directed mass spectrometry analysis, they found that TFIP11 is complexed with BLM. Further the authors implicate TFIP11 in cellular responses to replication stress, show its localization at stalled replication foci, determine the domain in TFIP11 which allows this to happen. Subsequently the authors try to determine mechanistically the function of TFIP11 at the stalled forks. In this section they do get confused about stalled forks and reversed forks, do not discriminate between the two terms and in fact uses them interchangeably without much justification. However at the end they do show that TFIP11 is essential for RAD51 recruitment and antagonises the effect of BLM at the stalled/reversed fork sites.

While the manuscript contains lots of data, most of which (with some exceptions listed below) are well controlled. What the manuscript lacks is proper referencing of the literature and trying to misappropriate known data as something which they have discovered. For example, the original references that had shown that BLM antagonizes RAD51 function in vivo have been missed (PMIDs 17591918, 17984114). The dual functions of BLM as a pro- and anti-recombinogenic protein (PMID 18003860) was not discussed in the context of both overexpression and ablation of BLM regulating RAD51 localization in PLA.

Major Comments:

1. The use of Mitomycin C is known to reduce the abundance of replication forks but not its progression (PMID: 25580447). Definitely its mechanism of action is different than that of HU. Hence it is suggested not to include the Mitomycin C data and instead concentrate on HU experiments. The Mitomycin C data does not give anything extra, instead causes more confusion.
2. Nowhere in the entire manuscript (except the Coomassies) are molecular weights given. All western blots should have molecular weights.
3. The initial iPOND experiment in Figure 3A should include known markers which are known to accumulate at the vicinity of stalled replication forks. For example the localization of BLM should be shown. In fact iPOND and mass spec analysis has shown that BLM is present at stalled replication sites (PMID 24047897). If this is indeed true – how does the authors reconcile their later results in Figures 6-8?

4. The authors should do PLA with BLM-RAD51, BLM-TFIP11, RAD51-TFIP11 combinations in +HU conditions. These PLA will give them more insight into the relationship between these three proteins.

5. Starting with Figure 5, the authors try to indicate that after HU treatment, fork stalling is same as fork reversal. Sometimes they also refer the phenomena as slowing of the fork. These are fundamentally not the same biochemical events, especially they use only once concentration of HU. Further their EM picture in Figure 5D is also not compelling regarding fork reversal as shown by multiple papers from Vindigni lab. A positive control (such as RECQ1) will help to conclusively prove the effect of the ablation of TFIP11 (Figure 5B). The authors must try to determine which biological feature they want to represent – stalled forks, slow forks or reversed forks and the entire manuscript should reflect that aspect. If they want to characterize the effect as reversed forks – the onus of proof rests with them.

6. The authors should explain how TFIP11 promotes replication fork reversal (Figure 5E-5F) and later TFIP11 wildtype was able to alleviate the defect in fork restart (Supplementary Figure 4B-4D)? Are these two not mechanistically separate events?

7. Figure 6D has to be supplemented with an iPOND experiment where BLM is depleted.

8. EM data should be shown for Figure 7G (in Supplementary Figures) where overexpression of BLM WT, decreased the levels of the reversed forks.

9. Figure 8F-8H – the authors have all the reagents to test the reverse effect – i.e. whether prebound TFIP11 was able to be displaced by increasing amount of recombinant BLM. This is essential to determine whether this is an unidirectional or bidirectional dynamic phenomena.

10. The authors should do a “washoff” experiments – treat with HU, wash it off and assay at two different time points (early and late) and try to see the recruitment of TFIP11, RAD51 and BLM using the biochemical (iPOND), DNA fibre assay and PLA. This will also give an idea about the interplay between the three main players.

Minor Comments:

1. All statistics should be done with Standard Deviation (SD) instead of Standard Error of Mean (SEM). Individual data points on histograms should be shown.

2. In Figure 2A – where are no deviations in cell viability?

3. Anti-Flag blot is necessary in Figure 3I.

4. Supplementary Figure 2A is incomplete and has limited value. Suddenly a splicing factor is brought in which has no relevance with the rest of the manuscript. In Supplementary Figure 2B, did the authors try to find the identity of the nuclear speckles? Is it PML NB – where BLM also resides in asynchronous cultures. Colocalization of TFIP11 with PML and BLM are necessary to complete Supplementary Figure 2B.

We would like to thank all the reviewers for their careful and constructive comments on our manuscript. Over the last three months, we have conducted a number of experiments as suggested by the reviewers. We hope that these new experimental findings address the reviewers' concerns and provide additional support for our main conclusions. Below is our point-by-point response to the reviewers' critiques:

Reviewer #1 (Remarks to the Author):

This manuscript from Chen et al. describes the identification of TFIP11, a splicing factor, as a binding partner of the Bloom's syndrome complex using co-immunoprecipitation techniques. Subsequently, the authors proceed to demonstrate that the depletion of TFIP11 via siRNAs leads to an increased susceptibility to agents inducing replication stress. Moreover, they observe notable modifications in the dynamics of replication forks and an elevated occurrence of chromosomal abnormalities. By employing iPOND and proximity ligation assays (PLAs), the authors assert that TFIP11 is recruited to replication forks upon exposure to hydroxyurea, potentially through direct DNA binding facilitated by its GCFC domain. Furthermore, they highlight the significance of this recruitment in TFIP11's role in promoting fork reversal. The authors end the manuscript by proposing that TFIP11 competes with BLM for access to stalled forks, thereby facilitating RAD51-mediated fork reversal. While the first part of the authors' manuscript about a role for TFIP11 in the replication stress response is largely convincing, the proposed mechanism involving BLM cannot be correct, as I explain below.

Thank you for the constructive suggestions and comments. We have now conducted several experiments to address the reviewer's comments and questions (please see details below).

Specific points:

1. The proposed hypothesis by the authors suggests a competitive relationship between TFIP11 and BLM regarding access to stalled forks. The authors suggest that TFIP11, by effectively out-competing BLM, hinders the removal of RAD51, subsequently promoting RAD51-dependent fork reversal. However, several issues arise in relation to this model. Firstly, according to the model, depletion of BLM should result in an increased occurrence of fork reversal. However, contrary to this expectation, the authors actually observe a reduction in the number of reversed forks in BLM-depleted cells, as depicted in Figure 7J. This observed outcome critically undermines the validity of the authors' proposed model.

We appreciate the reviewer for bringing up these important points.

Regarding the observed reduction in the frequency of reversed forks following HU

treatment upon BLM depletion, we acknowledge that the underlying reason remains unresolved. However, we have considered two plausible possibilities, as discussed in our manuscript. First, in the absence of BLM, it's possible that sustained RAD51 presence at stalled forks may impede the engagement of DNA translocases such as SMARCAL1, ZRANB3 or HLTF, potentially limiting the process of fork reversal. Second, given that RAD51 plays a crucial role in replication fork reversal in response to replication stress, the defective removal of RAD51 in BLM-depleted cells could potentially impair its recycling efficiency, ultimately resulting in a reduction in the frequency of reversed forks. These insights emphasize the critical importance of precisely regulating the activities of BLM/RAD51 at stalled forks, as either excessive or insufficient BLM/RAD51 activities can lead to impaired replication fork reversal, as discussed in our manuscript. We hope these explanations provide clarity on the points raised by the reviewer.

2. Secondly, according to the authors' proposed model, the loss of BLM should lead to an increased accumulation of RAD51 at stalled forks. However, previous studies, such as Bugreev et al. (JBC 2009) and Xue et al. (NAR 2019), have demonstrated that BLM is incapable of removing the active ATP-bound form of RAD51 from DNA. Additionally, studies by Patel et al. (JCB 2017), Shorrocks et al. (Nat. Comm. 2021), and others, have shown that the loss of BLM does not result in elevated RAD51 foci. It is likely that RECQL5, another RecQ family helicase, plays a much more significant role in countering RAD51 filament formation in human cells, as indicated by Hu et al. (Genes Dev. 2007) and Schwendener et al. (JBC 2010). The iPOND experiment presented in Figure 7D, which the authors claim demonstrates a 1.5-fold increase in RAD51 on nascent DNA in siBLM-treated cells, is not convincing. The observed increase is minimal, and the H3 loading control is over-exposed to the extent that assessing protein loading becomes impossible. Additionally, a similar experiment in Figure S6D involving overexpression of wild-type and helicase-dead BLM is depicted, but the level of BLM overexpression is so excessively high that its biological relevance becomes questionable when considering the extent of helicase overexpression.

We appreciate the reviewer for raising these important points.

In the studies by Patel et al. (JCB 2017) and Shorrocks et al. (Nat. Comm. 2021), quantification of RAD51 foci in wild-type and BLM-deficient cells was performed after treatment with 2 mM HU for 24 hours or exposure to ionizing radiation (IR). It's important to note that IR directly induces DNA double-strand breaks (DSBs), and treatment with 2 mM HU for an extended period can also lead to fork collapse and DSB formation. Consequently, in these contexts, RAD51 foci typically represent sites of ongoing DSB repair by homologous recombination (HR). In contrast, our study involved treating cells with HU for a shorter duration (3 hours). Importantly, we did not observe the activation of ATM, which is indicative of DSBs, in the absence or presence of HU treatment (Supplementary Figure 4B). This

suggests that in our experimental conditions, DSBs were not formed. Therefore, the RAD51/biotin PLA foci in our study specifically represent the presence of RAD51 at stalled forks, not DSBs or collapsed forks. This distinction implies that BLM might have the capacity to limit excessive RAD51 accumulation specifically at stalled forks, as discussed in our manuscript. Consistently, the studies, such as Bugreev et al. (JBC 2009) and Xue et al. (NAR 2019), which show that BLM is incapable of removing the active ATP-bound form of RAD51 from DNA in vitro, may also reflect the role of BLM at DSBs but not stalled forks.

Regarding the observed 1.5-fold increase in RAD51 on nascent DNA in BLM-depleted cells, we acknowledge that it may not appear as a dramatic change. However, considering the pivotal role of RAD51 in replication fork reversal, even a 1.5-fold change could have significant biological implications. We have repeated this experiment several times and consistently obtained the same results. More importantly, we have also observed a 1.5- to 2-fold increase in RAD51 on nascent DNA in RMI1-depleted cells (Supplementary Figure 6G). Additionally, we have adjusted the imaging conditions to ensure that the H3 loading control is not over-exposed (Figure 7D).

We also appreciate the reviewer's point about the excessively high levels of BLM overexpression. In response to this concern, we have reconstructed the expression plasmids for BLM to achieve expression levels that closely resemble endogenous BLM levels. As shown in the revised Figure S6E, the overexpression of wild-type BLM, but not the helicase-dead mutant, results in a reduction in the association of RAD51 with nascent DNA after replication stress.

We hope these explanations and the modifications in our experimental approach address the reviewer's concerns and provide a clearer understanding of our findings.

3. Thirdly, if the authors' model is correct, then why would BLM and TFIP11 need to interact at all if they compete for the same substrate? BLM is a common contaminant in proteomics experiments (see crapome.org), and TFIP11 seems to interact with thousands of cellular proteins. Can the authors really be sure the BLM-TFIP11 interaction is biologically relevant?

We appreciate the reviewer's thoughtful questions and concerns.

Regarding the interaction between TFIP11 and BLM and its biological relevance, we acknowledge the possibility that if both proteins compete for the same substrate, their interaction might seem counterintuitive. However, it's plausible that the interaction between TFIP11 and BLM induces a conformational change in BLM, which in turn facilitates its release from single-stranded DNA (ssDNA) at stalled forks. Therefore, TFIP11 could potentially antagonize BLM through two

mechanisms: (1) competition with BLM for binding to stalled forks; and (2) alteration of BLM's structural conformation. We believe these possibilities warrant further investigation, and as such, we have included them in the discussion within our manuscript. The proposed action model of TFIP11 and BLM at stalled forks is somewhat analogous to the action of RAD51 paralogues RAD55-RAD57 and the Srs2 helicase during HR repair, in which RAD55-RAD57 interacts with and inhibits Srs2 helicase¹.

Regarding the concern about BLM being a common contaminant in proteomics experiments, we appreciate this insight. It's important to note that in our experimental system, which involved a two-step purification procedure, we did not recognize BLM protein complexes as common contaminants. However, we acknowledge the reviewer's concern, and we will consider this aspect in our future experimental design and analysis.

4. Major conclusions in the manuscript heavily rely on a substantial number of PLA experiments. However, I find it perplexing that the authors have opted for this assay to demonstrate protein colocalization, considering its susceptibility to artifacts (as highlighted <https://www.biorxiv.org/content/10.1101/411355v2> for example). The fact that so few PLA foci are present in cells when there would be hundreds of stalled forks in a cell treated with hydroxyurea is problematic (e.g. Figure 4B). The authors should re-analyze by conventional IF microscopy the recruitment of proteins instead of using PLA.

We appreciate the concern raised by the reviewer regarding the use of PLA experiments for demonstrating protein localization at stalled replication forks. We acknowledge the potential susceptibility to artifacts associated with PLA, as highlighted in the provided reference.

We have now conducted siRNA experiments to validate the specificity of the antibodies used in our PLA experiments (Figure 6A-6C, Figure 8A-8C, and Supplementary Figure 8A-8C). This additional step confirms that the antibodies indeed measure colocalization of protein/DNA. In addition, in our manuscript, we have now used both iPOND and PLA experiments to comprehensively analyze the recruitment of TFIP11, RAD51, BLM, RMI1, and RPA2 to stalled forks (all PLA experiments, including those shown in Figure 4B, were accompanied by iPOND experiments, providing a parallel assessment of protein recruitment to stalled forks). These two different experimental approaches complement each other and have consistently supported our conclusions.

To address the reviewer's concern, we have also conducted several additional conventional immunostaining experiments. The results from these experiments reinforce and align with our original conclusions. If necessary, we will integrate them into the revised manuscript (Please see Figures below).

HeLa cells transfected with indicated siRNAs/plasmids were treated with 4 mM HU for 3 h. Cells were subjected to immunostaining using anti-RAD51 or anti-BLM antibodies. Representative images were shown (A, D, G). DNA was stained with DAPI. Scale bar, 10 μ m. Data represent means \pm SD from three independent experiments (B, E, H). More than 100 cells were counted in each experiment. ns, not significant; *** P <0.001, **** P <0.0001, one-way ANOVA test. Western blot analysis of TFIP11, RAD51, and BLM expression (C, F, I).

By combining PLA, iPOND, and conventional immunostaining experiments, we have taken a comprehensive approach to enhance the rigor and robustness of our conclusions regarding protein recruitment at stalled replication forks. We hope these clarifications address the concerns raised by the reviewer.

5. All western blots require molecular weight markers.

Thank you for your suggestion. Molecular weight markers have been included in all the revised figures.

Reviewer #2 (Remarks to the Author):

Chen and colleagues' manuscript is a well-written characterisation of a novel genome stability factor TFIP11. The overall conclusions are that TFIP11 promotes replication fork regression by preventing the dissociation of RAD51 from stalled forks by BLM helicase. The data as presented support the hypothesis: The findings are very interesting, although some weaknesses include:

Thank you for the constructive suggestions and comments. We have now conducted several experiments to address the reviewer's comments and questions (please see details below).

1. an over-reliance on proximity ligation assay experiments in support of conclusions, many times without the appropriate controls required for accurate reporting of PLA results (see for example <https://www.biorxiv.org/content/10.1101/411355v2.full>). The authors should indicate specifically which controls were used in all PLA experiments, and show that the antibodies that they are using in their experiments do indeed measure colocalization of proteins or protein/DNA, and not just changes in the levels of expression (eg siRNA, overexpression) of one partner relative to the other.

We greatly appreciate the concerns expressed by the reviewer regarding our use of PLA experiments to demonstrate protein localization at stalled replication forks and the potential susceptibility to artifacts, as highlighted in the provided reference. We fully recognize the critical importance of implementing appropriate controls in PLA experiments to ensure the accuracy and reliability of the results. In response to the reviewer's suggestion, we have conducted siRNA experiments to validate the specificity of the antibodies used in our PLA experiments (Figure 6A-6C, Figure 8A-8C, and Supplementary Figure 8A-8C). This additional step confirms that the antibodies indeed measure colocalization of protein/DNA.

In addition, in our manuscript, we have now used both iPOND and PLA experiments to comprehensively analyze the recruitment of TFIP11, RAD51, BLM, RMI1, and RPA2 to stalled forks (all PLA experiments were accompanied by iPOND experiments, providing a parallel assessment of protein recruitment). These two different experimental approaches complement each other and have consistently supported our conclusions.

Furthermore, we have conducted several additional conventional immunostaining experiments to further strengthen and corroborate our original findings. The results from these experiments reinforce and align with our original conclusions.

By combining PLA, iPOND, and conventional immunostaining experiments, we have taken a comprehensive approach to enhance the rigor and robustness of our conclusions regarding protein recruitment and colocalization at stalled replication

forks. We hope these clarifications address the concerns raised by the reviewer.

2. the use of bar graphs to present data that is likely hiding what should have a large spread of values. All the experiments measuring PLA numbers/nucleus, stalled forks and EM reversed forks should include the primary data points used to derive the mean and stdev. These graphs have such tight error bars, it is hard to know what “real values” were used to derive the results. The graphs should be redrawn showing the primary data points (similar style to 5H, I, J).

We appreciate your suggestion. We have now redrawn the graphs to include primary data points, following a similar style to Figure 5H, I, and J.

2a) One problem for presenting PLA foci “per positive cell” is that cells without PLA signals are not counted. Does the fraction of PLA positive/negative cells also change, or is it consistent (this is shown for 3D and 3H, but not other PLA graphs)? Showing a value for every nucleus (including those that are negative) is a much more open way of showing the data.

We appreciate the reviewer's suggestion. In our PLA experiments, cells with more than 2 PLA foci were considered positive to ensure that we focus on cells with a significant level of protein/DNA interaction. Including cells with fewer than 2 PLA foci may increase background noise and may not accurately reflect the true extent of interaction. However, we understand the importance of transparency in data presentation. To address this concern, we have now used both iPOND and PLA experiments to comprehensively analyze the recruitment of TFIP11, RAD51, BLM, RMI1, and RPA2 to stalled forks (all PLA experiments were accompanied by iPOND experiments, providing a parallel assessment of protein recruitment). These two different experimental approaches complement each other and have consistently supported our conclusions.

Furthermore, we have conducted several additional conventional immunostaining experiments to further strengthen and corroborate our original findings. The results from these experiments reinforce and align with our original conclusions.

2b) how is the standard deviation derived from the graphs Fig 5B, 5C, 7G, 7J? A number is given below each bar indicating the number of forks counted, but is this the total number of counts across three replicates, or the total number in each replicate?

We apologize for any confusion regarding the representation of standard deviation in the graphs (Figures 5B, 5C, 7G, 7J). The number provided below each bar in these graphs indicates the total number of counts across three replicates, not the total number in each replicate. The information has now been included in the revised manuscript.

3. Several experiments use siRNA+transient transfection experiments to show rescue of very sensitive results. The results as presented don't show what *percentage* of cells are transfected in the experiments. Complete rescue of the phenotype would require 100% of cells to be transfected, and expression to be relatively uniform across those cells. But I am skeptical that PEI can achieve 100% or even 50% in U2OS or HeLa cells where most experiments were done. But if this is possible it would be good to see the results, eg shown by FACS of a transfection reporter.

We apologize for any lack of clarity in the presentation of this information. In our experiments, for transfections conducted in HEK293T cells, we used polyethyleneimine following the manufacturer's guidelines. For transfections in U2OS or HeLa cells, we utilized Hieff Trans™ Liposomal Transfection Reagent following the manufacturer's guidelines (Yeasen Biotechnology). This information has now been included in the Materials and Methods section. We would like to clarify that our transfection efficiency exceeded 80%, as evidenced by immunostaining experiments (see Figures below).

TFIP11 is previously characterised for its role in splicing. Were splicing factors identified in the mass spectrometry experiment of Figure 1? Or were only the indicated DNA repair factors the only partner proteins identified? All interacting proteins should be indicated. The mass spec data should be deposited in the PRIDE repository so that it can be analysed by others.

Yes, several splicing factors were identified in the mass spectrometry experiment of Figure 1. Gene ontology (GO) term analysis revealed that TFIP11 interactors are highly involved in RNA splicing and telomere organization, which is consistent with previous studies ^{2,3,4,5} (Supplementary Figure 1A). Notably, we detected the presence of BLM and several established BLM-associated factors, including

TOP3A, RMI1, RMI2, RIF1, FANCI, and FANCD2, among the prominent TFIP11-associated proteins (Figure 1C and Supplementary Figure 1A). As suggested by the reviewer, the mass spec data has now been deposited in the PRIDE repository (Project accession: PXD042222). We appreciate the suggestion to make this data available for further analysis.

The authors conclude that TFIP11 blocks BLM activity on forks, and show that it competitively binds with BLM in a DNA pulldown experiments. This is a nice result, but one that could be more significant if an activity (helicase/unwinding) assay was performed with the proteins. This would conclusively demonstrate that TFIP11 is blocking the helicase activity of BLM.

Thank you for your insightful suggestion. We have now performed an in vitro DNA unwinding assay, which conclusively demonstrates that TFIP11 is able to inhibit the DNA unwinding ability of BLM in vitro (Supplementary Figure 9E). The results have been included in the revised manuscript.

There are many other translocase enzymes that can regress replication and/or restart forks (HLTF, SMARCAL1, ZRANB3, RECQ1, FANCM, etc). There should be some discussion about whether the authors think these enzymes could also be inhibited or promoted by binding of TFIP11 at the junction.

Thank you for your insightful suggestion. We agree that further studies should explore whether TFIP11 also plays a role in regulating other translocase enzymes involved in the regression and restart of replication forks, such as HLTF, SMARCAL1, ZRANB3, RECQ1, FANCM, among others. The potential interactions and regulatory mechanisms involving TFIP11 and these enzymes represent an interesting avenue for future research. We have now included a discussion of this possibility in the revised manuscript.

One minor issue to discuss: the authors show that TFIP11 knockdown leads to increased ssDNA (regressed forks), but they see no change in RPA2 foci formation. These results seem to conflict as most (all?) ssDNA is usually RPA coated in cells. Please comment on whether the ssDNA at regressed forks is bound by something other than RPA.

We apologize for the lack of clarity. The regressed arms of reversed replication forks exhibit structural similarities to one-ended double-stranded breaks, necessitating robust protection against uncontrolled nucleolytic degradation. Elegant research conducted by various laboratories^{6,7} has demonstrated that the stabilization of RAD51 at ssDNA of the regressed arms plays a critical role in protecting these structures. We hope this clarifies the issue raised by the reviewer.

The paper is well written and the function of TFIP11 described is novel. There is a lot of work done within the paper that supports the conclusions. Addressing the mostly

data-presentation related issues above will improve the robustness and reproducibility of the manuscript.

We appreciate the reviewer's positive feedback. We have taken into consideration the data presentation-related issues highlighted and made improvements to enhance the robustness and reproducibility of the manuscript. We hope these revisions have addressed the concerns effectively.

Reviewer #3 (Remarks to the Author):

Manuscript ID: NCOMMS-23-18765

Title: TFIP11 competes with the Bloom syndrome helicase for binding to stalled replication forks to preserve genome stability

Authors: Chen et al.,

In this manuscript the authors have tried to determine the role of GCFC domain-containing protein TFIP11 (also known as TIP39) after replication stress. Using a directed mass spectrometry analysis, they found that TFIP11 is complexed with BLM. Further the authors implicate TFIP11 in cellular responses to replication stress, show its localization at stalled replication foci, determine the domain in TFIP11 which allows this to happen. Subsequently the authors try to determine mechanistically the function of TFIP11 at the stalled forks. In this section they do get confused about stalled forks and reversed forks, do not discriminate between the two terms and in fact uses them interchangeably without much justification. However at the end they do show that TFIP11 is essential for RAD51 recruitment and antagonises the effect of BLM at the stalled/reversed fork sites.

While the manuscript contains lots of data, most of which (with some exceptions listed below) are well controlled. What the manuscript lacks is proper referencing of the literature and trying to misappropriate known data as something which they have discovered. For example, the original references that had shown that BLM antagonizes RAD51 function *in vivo* have been missed (PMIDs 17591918, 17984114). The dual functions of BLM as a pro- and anti-recombinogenic protein (PMID 18003860) was not discussed in the context of both overexpression and ablation of BLM regulating RAD51 localization in PLA.

Thank you for the concise summary and valuable suggestion. In the revised manuscript, we have addressed your concern by properly referencing the original studies that demonstrated the dual functions of BLM as both a pro- and anti-recombinogenic protein (PMID 18003860) and the role of BLM in antagonizing RAD51 function *in vivo* (PMIDs 17591918, 17984114) in both the Results and Discussion sections. We hope these additions enhance the clarity and accuracy of our manuscript regarding the context of BLM's functions. We also appreciate your feedback regarding the distinction between stalled forks, slow forks, and have made the necessary adjustments to improve the clarity and accuracy of our manuscript.

Major Comments:

1. The use of Mitomycin C is known to reduce the abundance of replication forks but not its progression (PMID: 25580447). Definitely its mechanism of action is different

than that of HU. Hence it is suggested not to include the Mitomycin C data and instead concentrate on HU experiments. The Mitomycin C data does not give anything extra, instead causes more confusion.

Thank you for the suggestion. We have now removed the Mitomycin C data from the revised manuscript, focusing solely on the HU experiments to avoid any potential confusion.

2. Nowhere in the entire manuscript (except the Coomassies) are molecular weights given. All western blots should have molecular weights.

Thank you for your suggestion. Molecular weight markers have been included in all the revised figures.

3. The initial iPOND experiment in Figure 3A should include known markers which are known to accumulate at the vicinity of stalled replication forks. For example the localization of BLM should be shown. In fact iPOND and mass spec analysis has shown that BLM is present at stalled replication sites (PMID 24047897). If this is indeed true – how does the authors reconcile their later results in Figures 6-8?

Thank you for your suggestion. We have now repeated the iPOND experiment in Figure 3A and included known markers including BLM, RMI1, and RAD51, which are known to accumulate at the vicinity of stalled replication forks. Our data indeed indicate that TFIP11, BLM, RMI1, and RAD51 accumulate at stalled forks upon replication stress. We speculate that TFIP11 and the BLM protein complex may compete with each other at the stalled forks and the precise balance and regulation of TFIP11 and BLM activities at stalled forks are critical for proper replication fork reversal and stability. Any disruption in this delicate balance could potentially lead to the impaired replication fork reversal observed in our later results in Figures 6-8.

Regarding the similar accumulation pattern and interaction of TFIP11 and BLM and their biological relevance, we acknowledge that if both proteins compete for the same substrate, their interaction might seem counterintuitive. However, it's plausible that the interaction between TFIP11 and BLM induces a conformational change in BLM, which in turn facilitates its release from single-stranded DNA (ssDNA) at stalled forks. Therefore, TFIP11 could potentially antagonize BLM through two mechanisms: (1) competition with BLM for binding to stalled forks; and (2) alteration of BLM's structural conformation. We believe these possibilities warrant further investigation, and as such, we have included them in the discussion within our manuscript. The proposed action model of TFIP11 and BLM at stalled forks is somewhat analogous to the action of RAD51 paralogues RAD55-RAD57 and the Srs2 helicase during HR repair, in which RAD55-RAD57 interacts with and inhibits Srs2 helicase¹.

4. The authors should do PLA with BLM-RAD51, BLM-TFIP11, RAD51-TFIP11 combinations in +HU conditions. These PLA will give them more insight into the relationship between these three proteins.

Thank you for your suggestion. We performed "washoff" experiments. Cells were treated with HU, followed by HU removal, and then assayed at various time points to investigate the accumulation of TFIP11, RAD51, and BLM at stalled forks using iPOND and PLA. As shown in Supplementary Figure 10, TFIP11, RAD51, and BLM all accumulated at stalled forks upon HU treatment. Intriguingly, after HU removal, all three proteins exhibited a similar pattern of dissociation from stalled forks over time. These findings further validate our initial observations of TFIP11 interacting with BLM and competing at stalled forks. This dynamic behavior of TFIP11, BLM, and RAD51 at stalled forks enhances our understanding of their interactions in response to replication stress.

5. Starting with Figure 5, the authors try to indicate that after HU treatment, fork stalling is same as fork reversal. Sometimes they also refer the phenomena as slowing of the fork. These are fundamentally not the same biochemical events, especially they use only once concentration of HU. Further their EM picture in Figure 5D is also not compelling regarding fork reversal as shown by multiple papers from Vindigni lab. A positive control (such as RECQ1) will help to conclusively prove the effect of the ablation of TFIP11 (Figure 5B). The authors must try to determine which biological feature they want to represent – stalled forks, slow forks or reversed forks and the entire manuscript should reflect that aspect. If they want to characterize the effect as reversed forks – the onus of proof rests with them.

Thank you for emphasizing the importance of distinguishing between stalled forks, slow forks, and reversed forks. Previous studies have shown that genetic defects in reversed fork formation upon genotoxic treatments impair active replication fork slowing, linking controlled fork progression and fork remodeling upon replication stress^{8, 9, 10}. Consistent with these findings, we found that inactivation of TFIP11 impairs replication stress-induced fork reversal, consequently resulting in impaired fork slowing. We have revised the manuscript to clarify these distinctions and have provided additional experimental data, including more compelling EM images and the inclusion of SMARCAL1 as a positive control, to support our characterization of the effect as reversed forks (Figure 5A-5B). Your feedback is valuable, and we have made the necessary adjustments to improve the clarity and accuracy of our manuscript.

6. The authors should explain how TFIP11 promotes replication fork reversal (Figure 5E-5F) and later TFIP11 wildtype was able to alleviate the defect in fork restart (Supplementary Figure 4B-4D)? Are these two not mechanistically separate events?

Thank you for pointing that out. Replication fork reversal is a crucial process that can bridge the stabilization and restart of stalled forks¹¹. It plays a pivotal role in promoting fork restart and maintaining fork integrity¹¹. We believe that the role of TFIP11 in promoting fork reversal is intricately linked to its function in facilitating stalled fork restart. We have revised the manuscript to provide a clearer explanation of this relationship between TFIP11's role in fork reversal and its impact on stalled fork restart.

7. Figure 6D has to be supplemented with an iPOND experiment where BLM is depleted.

We apologize for the lack of clarity. An iPOND experiment where BLM is depleted is indeed shown in Figure 7D.

8. EM data should be shown for Figure 7G (in Supplementary Figures) where overexpression of BLM WT, decreased the levels of the reversed forks.

Thank you for your suggestion. An electron micrograph of a representative replication fork and a reversed replication fork has been added to Supplementary Figure 7A in the revised manuscript.

9. Figure 8F-8H – the authors have all the reagents to test the reverse effect – i.e. whether prebound TFIP11 was able to be displaced by increasing amount of recombinant BLM. This is essential to determine whether this is an unidirectional or bidirectional dynamic phenomena.

Thank you for your suggestion. We have conducted the suggested experiments and found that pre-bound TFIP11 can also be displaced by increasing amount of recombinant BLM. The data has now been included in the revised manuscript (Supplementary Figure 9G).

10. The authors should do a “washoff” experiments – treat with HU, wash it off and assay at two different time points (early and late) and try to see the recruitment of TFIP11, RAD51 and BLM using the biochemical (iPOND), DNA fibre assay and PLA. This will also give an idea about the interplay between the three main players.

As per the reviewer's suggestion, we conducted "washoff" experiments by treating cells with HU, washing it off, and assaying at different time points to examine the accumulation of TFIP11, RAD51, and BLM at stalled forks using iPOND and PLA. As shown in Supplementary Figure 10, TFIP11, RAD51, and BLM all accumulate at stalled forks upon HU treatment. Interestingly, after HU removal, all three proteins quickly dissociated from stalled forks in a similar time pattern. These results further support our initial findings that TFIP11 interacts with BLM and compete with each other at stalled forks. These findings highlight the dynamic

nature of TFIP11, BLM, and RAD51 at stalled forks and provide a more comprehensive understanding of their interactions in response to replication stress.

Minor Comments:

1. All statistics should be done with Standard Deviation (SD) instead of Standard Error of Mean (SEM). Individual data points on histograms should be shown.

Thank you for your suggestion. We have now used Standard Deviation (SD) for all statistics instead of Standard Error of Mean (SEM), and individual data points on histograms have been included in the revised manuscript.

2. In Figure 2A – where are no deviations in cell viability?

Thank you for your suggestion. Deviations have now been included in revised Figure 2A.

3. Anti-Flag blot is necessary in Figure 3I.

Thank you for your suggestion. An anti-Flag blot has been included in the revised Figure 3I as recommended.

4. Supplementary Figure 2A is incomplete and has limited value. Suddenly a splicing factor is brought in which has no relevance with the rest of the manuscript. In Supplementary Figure 2B, did the authors try to find the identity of the nuclear speckles? Is it PML NB – where BLM also resides in asynchronous cultures. Colocalization of TFIP11 with PML and BLM are necessary to complete Supplementary Figure 2B.

We apologize for the lack of clarity. TFIP11 was previously identified as a splicing factor^{2, 3, 4, 5}. In Supplementary Figure 2A, we performed immunostaining experiments and found that TFIP11 indeed partially colocalizes with splicing factor snRNP70. Interestingly, upon replication stress, a small fraction of TFIP11 is re-localized to stalled forks where it colocalizes with RPA (RPA has been shown to colocalize with BLM at stalled forks¹²) (Supplementary Figure 2B). We have now described the published roles of TFIP11 in RNA splicing in the revised manuscript.

1. Liu J, Renault L, Veaute X, Fabre F, Stahlberg H, Heyer WD. Rad51 paralogues Rad55-Rad57 balance the antirecombinase Srs2 in Rad51 filament formation. *Nature* **479**, 245-248 (2011).
2. Zhou Z, Licklider LJ, Gygi SP, Reed R. Comprehensive proteomic analysis of the human spliceosome. *Nature* **419**, 182-185 (2002).

3. Wen X, Lei YP, Zhou YL, Okamoto CT, Snead ML, Paine ML. Structural organization and cellular localization of tuftelin-interacting protein 11 (TFIP11). *Cell Mol Life Sci* **62**, 1038-1046 (2005).
4. Duchemin A, *et al.* DHX15-independent roles for TFIP11 in U6 snRNA modification, U4/U6.U5 tri-snRNP assembly and pre-mRNA splicing fidelity. *Nat Commun* **12**, 6648 (2021).
5. Herrmann G, *et al.* Conserved interactions of the splicing factor Ntr1/Spp382 with proteins involved in DNA double-strand break repair and telomere metabolism. *Nucleic Acids Res* **35**, 2321-2332 (2007).
6. Schlacher K, Christ N, Siaud N, Egashira A, Wu H, Jasin M. Double-strand break repair-independent role for BRCA2 in blocking stalled replication fork degradation by MRE11. *Cell* **145**, 529-542 (2011).
7. Taglialatela A, *et al.* Restoration of Replication Fork Stability in BRCA1- and BRCA2-Deficient Cells by Inactivation of SNF2-Family Fork Remodelers. *Mol Cell* **68**, 414-430 e418 (2017).
8. Vujanovic M, *et al.* Replication Fork Slowing and Reversal upon DNA Damage Require PCNA Polyubiquitination and ZRANB3 DNA Translocase Activity. *Mol Cell* **67**, 882-890 e885 (2017).
9. Kile AC, *et al.* HLTf's Ancient HIRAN Domain Binds 3' DNA Ends to Drive Replication Fork Reversal. *Molecular cell* **58**, 1090-1100 (2015).
10. Ray Chaudhuri A, *et al.* Topoisomerase I poisoning results in PARP-mediated replication fork reversal. *Nat Struct Mol Biol* **19**, 417-423 (2012).
11. Liao H, Ji F, Helleday T, Ying S. Mechanisms for stalled replication fork stabilization: new targets for synthetic lethality strategies in cancer treatments. *EMBO Rep* **19**, (2018).
12. Shorrocks AK, *et al.* The Bloom syndrome complex senses RPA-coated single-stranded DNA to restart stalled replication forks. *Nat Commun* **12**, 585 (2021).

REVIEWER COMMENTS

Reviewer #1 (Remarks to the Author):

I had four major points of concern in my original review. Point 4 was a concern about PLA, which the authors have addressed by carrying out orthogonal iPOND experiments. However, they have addressed my first three points with little more than conjecture. Thus my original concerns remain.

As I said in my previous review, I think most of Figures 1-6 are convincing but Figure 7-8 and the proposed mechanism regarding TFIP11 counteracting BLM to promote RAD51 loading, is problematic. This is because 1) the authors' own data shows that reversed forks in BLM-deficient cells are reduced rather than increased as their model would predict; and 2) BLM is known not to have a role in counteracting RAD51 loading, either in vitro or in cells (Bugreev et al. JBC 2009, Xue et al. NAR 2019, Patel et al. (JCB 2017), Shorrocks et al. (Nat. Comm. 2021)).

I therefore still cannot recommend publication unfortunately.

Reviewer #2 (Remarks to the Author):

This is a far superior manuscript to that originally submitted. The authors have done an excellent job addressing my original concerns, and my fellow reviewers. This is an excellent piece of finished work.

Reviewer #3 (Remarks to the Author):

Manuscript ID: NCOMMS-23-18765

Title: TFIP11 competes with the Bloom syndrome helicase for binding to stalled replication forks to preserve genome stability

Authors: Chen et al.,

In the revised version of the manuscript the authors have done most of the experiments which had been suggested by the reviewers. In a few cases appropriate explanations have been provided. The only part which the authors have to look and modify the discussion is listed below:

According to the iPOND experiment in Figure 3A, the new Supplementary Figure 9G and as also elaborated a few times in the rebuttal letter - the authors agree that the relationship between TFIP11 and BLM at the reversed forks is bi-directional and probably a product of built-in cellular redundancy. However throughout the entire manuscript the authors promote the idea that TFIP11 antagonises BLM at the forks. The authors need to correct this perception throughout the manuscript - especially in the abstract and the discussion sections - which will bring about a more balanced view about what is probably happening in the cellular context. Specifically the authors should discuss about the most probable dynamic nature of the interactions between TFIP11 and BLM at the forks under the different physiological contexts.

Point-by-point response to reviewers' comments:

Reviewer #1 (Remarks to the Author):

I had four major points of concern in my original review. Point 4 was a concern about PLA, which the authors have addressed by carrying out orthogonal iPOND experiments. However, they have addressed my first three points with little more than conjecture. Thus my original concerns remain.

As I said in my previous review, I think most of Figures 1-6 are convincing but Figure 7-8 and the proposed mechanism regarding TFIP11 counteracting BLM to promote RAD51 loading, is problematic. This is because 1) the authors own data shows that reversed forks in BLM-deficient cells are reduced rather than increased as their model would predict; and 2) BLM is known not to have a role in counteracting RAD51 loading, either in vitro or in cells (Bugreev et al. JBC 2009, Xue et al. NAR 2019, Patel et al. (JCB 2017), Shorrocks et al. (Nat. Comm. 2021)).

I therefore still cannot recommend publication unfortunately.

There may be a potential misunderstanding by Reviewer #1 regarding the roles of the RAD51 recombinase in HR-mediated DSB repair and replication fork reversal upon replication stress. To bring clarity to these roles, it is crucial to distinguish that RAD51 serves at least three distinct functions:

1) RAD51 in HR: RAD51 is an essential factor in HR and plays crucial roles in the repair of DSBs; 2) Protection of Stalled Replication Forks: RAD51 is also involved in safeguarding stalled replication forks against nuclease-catalyzed degradation of nascent DNA strands; 3) Replication Fork Reversal: In addition to its roles in HR and fork protection, RAD51 plays a role in promoting replication fork reversal in response to replication stress.

It's important to note that, while the HR and fork protection functions of RAD51 are strictly dependent on the mediator protein BRCA2, which assists RAD51 in replacing RPA to form a stable nucleofilament, the fork reversal function of RAD51 operates independently of BRCA2, and the formation of stable RAD51 nucleofilaments is not a prerequisite for replication fork reversal¹. In line with this, replication stress-induced RAD51 recruitment remains intact in BRCA2-deficient cancer cells and BRCA2-hypomorphic mutant embryonic stem cells^{2,3}.

As pointed out by Reviewer #1, previous studies have shown that the ability of BLM to disrupt the hRad51-ssDNA filament depends on the conformation of the filament: BLM can only disrupt an inactive filament in an ADP-bound form⁴. In contrast, BLM does not act upon the active ATP-bound form of the RAD51 filaments⁴. Consequently, during the HR process, where stable RAD51 filaments are formed with the assistance of BRCA2, BLM does not act upon the active ATP-

bound form of the RAD51 filaments. Given that the fork reversal function of RAD51 operates independently of BRCA2 and stable RAD51 nucleofilaments, it is reasonable that BLM can counteract RAD51's activities during the fork reversal process in response to replication stress. This highlights the intricate nature of RAD51's functions in different cellular contexts.

In the studies by Patel et al. (JCB 2017) and Shorrocks et al. (Nat. Comm. 2021), quantification of RAD51 foci in wild-type and BLM-deficient cells was performed after treatment with 2 mM HU for 24 hours or exposure to ionizing radiation (IR). It's important to note that IR directly induces DNA double-strand breaks (DSBs), and treatment with 2 mM HU for an extended period can also lead to fork collapse and DSB formation. Consequently, in these contexts, RAD51 foci typically represent sites of ongoing DSB repair by HR. In contrast, our study involved treating cells with HU for a shorter duration (3 hours). Importantly, we did not observe the activation of ATM, which is indicative of DSBs, in the absence or presence of HU treatment (Supplementary Figure 4B). This suggests that in our experimental conditions, DSBs were not formed. Therefore, the RAD51/biotin PLA foci in our study specifically represent the presence of RAD51 at stalled forks, not DSBs or collapsed forks. This distinction implies that BLM might have the capacity to limit excessive RAD51 accumulation specifically at stalled forks, as discussed in our manuscript. Consistently, the studies, such as Bugreev et al. (JBC 2009) and Xue et al. (NAR 2019), which show that BLM is incapable of removing the active ATP-bound form of RAD51 from DNA *in vitro*, may also reflect the role of BLM at DSBs but not stalled forks.

In response to Reviewer #1's comment regarding reversed forks in BLM-deficient cells are reduced rather than increased, as we discussed in our revised manuscript, it is not uncommon in the field to recognize that the intricate dynamics governing the assembly and disassembly of DNA damage/replication stress response factors at sites of DNA damage/stalled forks are crucial for a prompt response to DNA damage/replication stress. For example, both excessive and insufficient activities of MDC1 or CtIP at DSBs have been shown to impair HR repair (Han et al., 2021; Luo et al., 2012). This illustrates the complexity of the regulatory mechanisms involved in the response to DNA damage/replication stress.

As discussed in our manuscript, we have considered two plausible possibilities. First, in the absence of BLM, it's possible that sustained RAD51 presence at stalled forks may impede the engagement of DNA translocases such as SMARCAL1, ZRANB3 or HLTF, potentially limiting the process of fork reversal. Second, given that RAD51 plays a crucial role in replication fork reversal in response to replication stress, the defective removal of RAD51 in BLM-depleted cells could potentially impair its recycling efficiency, ultimately resulting in a reduction in the frequency of reversed forks.

Reviewer #2 (Remarks to the Author):

This is a far superior manuscript to that originally submitted. The authors have done an excellent job addressing my original concerns, and my fellow reviewers. This is an excellent piece of finished work.

Thanks!

Reviewer #3 (Remarks to the Author):

Manuscript ID: NCOMMS-23-18765

Title: TFIP11 competes with the Bloom syndrome helicase for binding to stalled replication forks to preserve genome stability

Authors: Chen et al., In the revised version of the manuscript the authors have done most of the experiments which had been suggested by the reviewers. In a few cases appropriate explanations have been provided. The only part which the authors have to look and modify the discussion is listed below:

According to the iPOND experiment in Figure 3A, the new Supplementary Figure 9G and as also elaborated a few times in the rebuttal letter - the authors agree that the relationship between TFIP11 and BLM at the reversed forks is bi-directional and probably a product of built-in cellular redundancy. However throughout the entire manuscript the authors promote the idea that TFIP11 antagonises BLM at the forks. The authors need to correct this perception throughout the manuscript - especially in the abstract and the discussion sections - which will bring about a more balanced view about what is probably happening in the cellular context. Specifically the authors should discuss about the most probable dynamic nature of the interactions between TFIP11 and BLM at the forks under the different physiological contexts.

Thank you for the constructive suggestions and comments. We have made the necessary revisions in the manuscript to incorporate this insightful recommendation.

1. Mijic S, *et al.* Replication fork reversal triggers fork degradation in BRCA2-defective cells. *Nat Commun* **8**, 859 (2017).
2. Ray Chaudhuri A, *et al.* Replication fork stability confers chemoresistance in BRCA-deficient cells. *Nature* **535**, 382-387 (2016).
3. Tarsounas M, Davies D, West SC. BRCA2-dependent and independent

formation of RAD51 nuclear foci. *Oncogene* **22**, 1115-1123 (2003).

4. Bugreev DV, Yu X, Egelman EH, Mazin AV. Novel pro- and anti-recombination activities of the Bloom's syndrome helicase. *Genes Dev* **21**, 3085-3094 (2007).

REVIEWERS' COMMENTS

Reviewer #4 (Remarks to the Author):

MS 424899 Chen et al. TFIP11 competes with the Bloom syndrome helicase for binding to stalled replication forks to preserve genome stability

This manuscript describes interactions between TFIP11 and the BLM-TOP3a-RMI1-RMI2 (BTRR) complex during replication stress generated by three-hour treatments with hydroxyurea (HU) mostly in HeLa cells. The work shows that TFIP11 is important in fork stability and restart; it interacts with BTRR and influences the accumulations of RAD51 whilst having no apparent effects on RPA accumulations. There are many interesting observations in the manuscript about TFIP11 and its effects on replication stress. The work is in many ways a tour de force, marshalling multiple high-end techniques to generate results, including mass spectrometry, DNA combing, iPOND, DNA electron microscopy, protein-DNA binding experiments, and immunofluorescence methods.

In my view, the results are a solid foundation from which to argue their points. There is a lot of reliance on the proximity ligation assay (PLA). It would help reader comprehension of the manuscript if the authors used their standard immunofluorescence data in Supplementary Figure 2 with RPA2 and TFIP11 and their RPA2/biotin data in Supplementary Figure 5 to benchmark the PLA to standard IF. It seems the treatment-group threshold the authors used squashes the number of foci down to about 10 in the majority of their experiments, whereas the numbers of foci generated by replication stress one finds in the literature fluctuates between 30 and 50 with large variances across positive cells, at least with a rapidly dividing cell line like HeLa. I am having trouble reconciling the fact that TFIP11 interacts with a "small fraction" (the authors' words) of RPA2-marked stalled forks after 3-h HU treatment with the size of the effects they see on BLM and RAD51 foci, considering the high percentages of co-localization of these factors after replication stress. The benchmarking the PLA could be considered in the discussion. If these PLA foci are the biggest and brightest foci of the pack, then perhaps they have the most forks in them and have the largest impact in the measurements. If the PLA foci numbers are not an effect of thresholding the image data, then it might be necessary to dig deeper into why the numbers are so different. In general, PLA is said to be more sensitive and one should be able to detect even lone forks. These data are not that.

The resection assay is over-interpreted. The authors did not see an increase in fork resection; however, if there is strand-specific nucleolytic resection (leading but not lagging, or visa versa), then the combing will show no change when resection is in fact occurring. Consequently, saying that the ssDNA all comes from the R-arm of reserved forks is questionable.

In the discussion starting at line 366, the authors could add the results with SMC5/6 and SUMO to this list of factors that affect the accumulations of RAD51 and RPA at stalled forks. Given the numbers of factors that affect RAD51 and RPA at stalled forks, I would say it is hard to conclude that there is direct competition between BLM and TFIP11 for stalled fork substrate. Most of the experiments were performed after three hours of HU treatment, while direct competition would be something that happens with minutes of fork stalling and the testing of the proposed mechanism would require dynamic measurement taken over those early timepoints. The evidence for BLM helicase specifically operating in fork reversal is not supported by the evidence here or in the literature. BLM levels at the fork affect RAD51 levels at the fork which could in turn affect fork remodeling and reversal. You can rewrite that sentence with other factors that influence RAD51, and then include that possibility these factors influence each other in vivo. BTRR and TFIP11 do not need to interact directly or compete for the same substrate for these considerations to apply.

I recommend that the authors modify their title and abstract and soften their conclusions with regards to direct competition between BTRR and TFIP11. They are at liberty to propose the competition model

as a potential explanation for their results and test the hypothesis in future studies. There is plenty of good stuff in the manuscript without going as far as they went.

Point-by-point response to reviewers' comments:

Reviewer #4 (Remarks to the Author):

MS 424899 Chen et al. TFIP11 competes with the Bloom syndrome helicase for binding to stalled replication forks to preserve genome stability

This manuscript describes interactions between TFIP11 and the BLM-TOP3a-RMI1-RMI2 (BTRR) complex during replication stress generated by three-hour treatments with hydroxyurea (HU) mostly in HeLa cells. The work shows that TFIP11 is important in fork stability and restart; it interacts with BTRR and influences the accumulations of RAD51 whilst having no apparent effects on RPA accumulations. There are many interesting observations in the manuscript about TFIP11 and its effects on replication stress. The work is in many ways a tour de force, marshalling multiple high-end techniques to generate results, including mass spectrometry, DNA combing, iPOND, DNA electron microscopy, protein-DNA binding experiments, and immunofluorescence methods.

Thanks for the nice summary!

In my view, the results are a solid foundation from which to argue their points. There is a lot of reliance on the proximity ligation assay (PLA). It would help reader comprehension of the manuscript if the authors used their standard immunofluorescence data in Supplementary Figure 2 with RPA2 and TFIP11 and their RPA2/biotin data in Supplementary Figure 5 to benchmark the PLA to standard IF. It seems the treatment-group threshold the authors used squashes the number of foci down to about 10 in the majority of their experiments, whereas the numbers of foci generated by replication stress one finds in the literature fluctuates between 30 and 50 with large variances across positive cells, at least with a rapidly dividing cell line like HeLa. I am having trouble reconciling the fact that TFIP11 interacts with a “small fraction” (the authors' words) of RPA2-marked stalled forks after 3-h HU treatment with the size of the effects they see on BLM and RAD51 foci, considering the high percentages of co-localization of these factors after replication stress. The benchmarking the PLA could be considered in the discussion. If these PLA foci are the biggest and brightest foci of the pack, then perhaps they have the most forks in them and have the largest impact in the measurements. If the PLA foci numbers are not an effect of thresholding the image data, then it might be necessary to dig deeper into why the numbers are so different. In general, PLA is said to be more sensitive and one should be able to detect even lone forks. These data are not that.

We appreciate the reviewer's observation and concern regarding the number of foci generated in our experiments. In response to this, we would like to clarify that the treatment-group threshold we employed was intentional and aimed at ensuring a reliable and specific signal in our PLA experiments. Specifically, we opted for a lower

concentration of antibodies during the PLA experiments to minimize background noise and enhance the specificity of the signal.

The resection assay is over-interpreted. The authors did not see an increase in fork resection; however, if there is strand-specific nucleolytic resection (leading but not lagging, or *visa versa*), then the combing will show no change when resection is in fact occurring. Consequently, saying that the ssDNA all comes from the R-arm of reserved forks is questionable.

Thank you for the feedback. We acknowledge the reviewer's concerns regarding the limitations of the native BrdU assay for detecting fork reversal *in vivo*. In our manuscript, we have employed a variety of approaches, including the native BrdU assay, DNA fiber assay, and electron microscopy, to comprehensively investigate the role of TFIP11 in fork reversal. These complementary techniques collectively contribute to a thorough understanding of TFIP11's impact on replication fork dynamics.

In the discussion starting at line 366, the authors could add the results with SMC5/6 and SUMO to this list of factors that affect the accumulations of RAD51 and RPA at stalled forks. Given the numbers of factors that affect RAD51 and RPA at stalled forks, I would say it is hard to conclude that there is direct competition between BLM and TFIP11 for stalled fork substrate. Most of the experiments were performed after three hours of HU treatment, while direct competition would be something that happens with minutes of fork stalling and the testing of the proposed mechanism would require dynamic measurement taken over those early timepoints. The evidence for BLM helicase specifically operating in fork reversal is not supported by the evidence here or in the literature. BLM levels at the fork affect RAD51 levels at the fork which could in turn affect fork remodeling and reversal. You can rewrite that sentence with other factors that influence RAD51, and then include that possibility these factors influence each other *in vivo*. BTRR and TFIP11 do not need to interact directly or compete for the same substrate for these considerations to apply.

I recommend that the authors modify their title and abstract and soften their conclusions with regards to direct competition between BTRR and TFIP11. They are at liberty to propose the competition model as a potential explanation for their results and test the hypothesis in future studies. There is plenty of good stuff in the manuscript without going as far as they went.

Thank you for the constructive suggestions and comments. In response to the reviewer's feedback, we have made significant revisions to our manuscript. Specifically, we have modified the title from "TFIP11 competes with the Bloom syndrome helicase for binding to stalled replication forks to preserve genome stability" to "TFIP11 collaborates with the

Bloom syndrome helicase to modulate RAD51 activity at stalled replication forks and promote fork reversal.”

Additionally, in the revised Abstract, we refrained from explicitly stating a direct competition between BTRR and TFIP11. Instead, we revised our statement to convey that “In this study, we demonstrate that the GCFC domain-containing protein TFIP11 forms a complex with the BLM helicase. TFIP11 exhibits a preference for binding to DNA substrates that mimic the structure generated at stalled replication forks.”

Further adjustments were made in the Introduction section, where we modified the statement from “In the present study, we demonstrate that TFIP11 plays a critical role in replication fork reversal by competing with BLM for binding to stalled replication forks, thereby controlling the activities of BLM and RAD51 at perturbed replication forks” to “In this study, we reveal that TFIP11 plays a crucial role in replication fork reversal by modulating the activities of BLM and RAD51 at stalled forks.”

Moreover, in the last paragraph of the Results section, we changed the subtitle from “TFIP11 and BLM compete for localization at stalled forks” to “TFIP11 and BLM exhibit counteractive dynamics at stalled forks.”

We believe that these changes align with the reviewer's recommendations and enhance the clarity and accuracy of our findings.

Additionally, we introduced a discussion on the roles of SMC5/6 and SUMO, highlighting their impact on RAD51 and RPA accumulations at stalled forks, with appropriate citation (Reference 73).